# An In-Context Learning Agent for Formal Theorem-Proving

**Amitayush Thakur, George Tsoukalas, Yeming Wen, Jimmy Xin & Swarat Chaudhuri**
Department of Computer Science
The University of Texas at Austin
{amitayush, george.tsoukalas, ywen, jxin31415}@utexas.edu, swarat@cs.utexas.edu

## Abstract

We present an in-context learning agent for formal theorem-proving in environments like Lean and Coq. Current state-of-the-art models for the problem are finetuned on environment-specific proof data. By contrast, our approach, called COPRA, repeatedly asks a high-capacity, general-purpose large language model (GPT-4) to propose tactic applications from within a stateful backtracking search. Proposed tactics are executed in the underlying proof environment. Feedback from the execution is used to build the prompt for the next model query, along with selected information from the search history and lemmas retrieved from an external database. We evaluate our implementation of COPRA on the `miniF2F` benchmark for Lean and a set of Coq tasks from the CompCert project. On these benchmarks, COPRA significantly outperforms few-shot invocations of GPT-4. It also compares favorably against finetuning-based approaches, outperforming REPROVER, a state-of-the-art finetuned approach for Lean, in terms of the *pass*@1 metric. Our code and data are available at https://github.com/trishullab/copra.

## 1 Introduction

The ability of large language models (LLMs) to learn in-context (Brown et al., 2020) is among the most remarkable recent findings in machine learning. Since its introduction, in-context learning has proven fruitful in a broad range of domains, from text and code generation (OpenAI, 2023b) to image generation (Ramesh et al., 2021) to game-playing (Wang et al., 2023a). In many tasks, it is now best practice to prompt a general-purpose, black-box LLM rather than finetune smaller-scale models.

In this paper, we investigate the value of in-context learning in the discovery of formal proofs written in frameworks like Lean (de Moura et al., 2015) and Coq (Huet et al., 1997). Such frameworks allow proof goals to be iteratively simplified using *tactics* such as variable substitution and induction. A *proof* is a sequence of tactics that iteratively "discharges" a given goal.

Automatic theorem-proving is a longstanding challenge in computer science (Newell et al., 1957). Traditional approaches to the problem were based on discrete search and had difficulty scaling to complex proofs (Bundy, 1988; Andrews & Brown, 2006; Blanchette et al., 2011). More recent work has used neural models — most notably, autoregressive language models (Polu & Sutskever, 2020; Han et al., 2021; Yang et al., 2023) — that generate a proof tactic by tactic.

The current crop of such models is either trained from scratch or fine-tuned on formal proofs written in a specific proof framework. By contrast, our method uses a highly capable, general-purpose, black-box LLM (GPT-4-turbo (OpenAI, 2023a) * ) that can learn in context. We show that few-shot prompting of GPT-4 is not effective at proof generation. However, one can achieve far better performance with an *in-context learning agent* (Yao et al., 2022; Wang et al., 2023a; Shinn et al., 2023) that repeatedly invokes GPT-4 from within a higher-level backtracking search and uses retrieved knowledge and rich feedback from the proof

---

*For brevity, we refer to GPT-4-turbo as GPT-4 in the rest of this paper.

environment. Without any framework-specific training, our agent achieves performance comparable to — and by some measures better than — state-of-the-art finetuned approaches.

Figure 1 gives an overview of our agent, called COPRA. [†] The agent takes as input a formal statement of a theorem and optional natural-language hints about how to prove the theorem. At each time step, it prompts the LLM to select the next tactic to apply. LLM-selected tactics are "executed" in the underlying proof assistant. If the tactic fails, COPRA records this information and uses it to avoid future failures.

Additionally, the agent uses lemmas and definitions retrieved from an external database to simplify proofs. Finally, we use a symbolic procedure (Sanchez-Stern et al., 2020) to only apply LLM-selected tactics when they simplify the current proof goals (ruling out, among other things, cyclic tactic sequences).

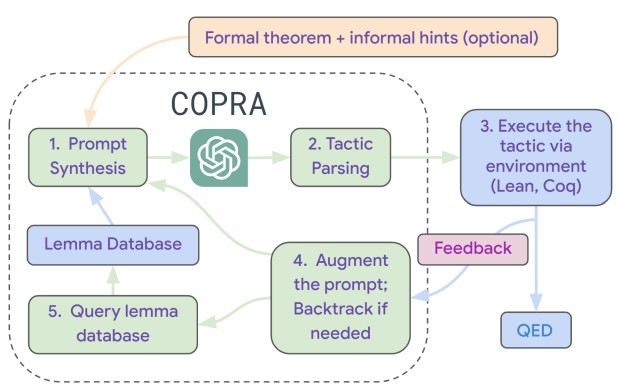

Figure 1: Overview of COPRA. COPRA takes as input a formal theorem statement and performs a stateful proof search through an in-context learning agent. At each step of search, the agent has access to the search history, error feedback from the proof environment, and relevant lemmas retrieved from a database, which are formatted according to a prompt serialization protocol.

Our use of a general-purpose LLM makes it easy to integrate our approach with different proof environments. Our current implementation of COPRA allows for proof generation in both Lean and Coq. To our knowledge, this implementation is the first proof generation system with this capability.

We evaluate COPRA using the `miniF2F` (Zheng et al., 2021) benchmark for competition-level mathematical reasoning in Lean, as well as a set of Coq proof tasks (Sanchez-Stern et al., 2020) from the CompCert (Leroy, 2009) project on verified compilation. In both settings, COPRA outperforms few-shot calls to GPT-4. On `miniF2F`, COPRA outperforms REPROVER, a state-of-the-art finetuned approach for Lean theorem-proving, in terms of the established *pass@1* metric. Using a refinement of the *pass@1* metric, we show that COPRA can converge to correct proofs with fewer model queries than REPROVER as well as PROVERBOT9001, a state-of-the-art approach for Coq. Finally, we show that a GPT-4-scale model is critical for this setting; an instantiation of COPRA with GPT-3.5 is less effective.

To summarize our contributions, we offer: (i) a new approach to formal theorem-proving, called COPRA, that combines in-context learning with search, knowledge retrieval, and the use of rich feedback from the underlying proof environment; (ii) a systematic study of the effectiveness of the GPT-4-based instantiation of COPRA, compared to few-shot GPT-4 invocations, ablations based on GPT-3.5, and state-of-the-art finetuned baselines; (iii) an implementation of COPRA, which is the first open-source theorem-proving system to be integrated with both the Lean and Coq proof environments.

## 2  Problem Formulation

A *(tactic-based) theorem-prover* starts with a set of unmet *proof obligations* and applies a sequence of *proof tactics* to progressively eliminate these obligations. Each obligation $o$ is a pair $(g, h)$, where $g$ is a *goal* and $h$ is a *hypothesis*. The goal $g$ consists of the propositions that need to be proved in order to meet $o$; the hypothesis $h$ is the set of assumptions that can be made in the proof of $g$. The prover seeks to reduce the obligation set to the empty set.

---

[†]COPRA is an acronym for "In-**co**ntext **Pr**over **A**gent".

Figure 2-(a) shows a formal proof, in the Lean language (de Moura et al., 2015), of a basic theorem about modular arithmetic (the proof is automatically generated using COPRA). The first step of the proof is the `intro` tactic, which changes a goal $P \rightarrow Q$ to a hypothesis $P$ and a goal $Q$. The next few steps use the `rw` (rewrite) tactic, which applies substitutions to goals and hypotheses. The last step is an application of the `refl` (reflexivity) tactic, which asserts definitional equivalences.

In code generation settings, LLMs like GPT-4 can often generate complex programs from one-shot queries. However, one-shot queries to GPT-4 often fail at even relatively simple formal proof tasks.

Figure 2-(c) shows a GPT-4-generated incorrect Lean proof of our example theorem.

By contrast, we follow a classic view of theorem-proving as a discrete search problem (Bundy, 1988). Specifically, we offer an agent that searches the state space of the underlying proof environment and discovers sequences of tactic applications that constitute proofs. The main differences from classical symbolic approaches are that the search is LLM-guided, history-dependent, and can use natural-language insights provided by the user or the environment.

```
(a)                       (c)

theorem                   begin
    mod_arith_2             intro h,
(x : ℕ) : x % 2            have h1 : x = 2
    = 0                         * (x / 2)
    → (x * x) % 2              := (nat.
    = 0 :=                      mul_div_cancel
begin                           ' h).symm,
  intro h,                    rw h1,
  rw mul_mod,                 rw nat.
  rw h,                         mul_div_assoc
  rw zero_mul,
  refl,                       -
end                           (show 2 | 2,
                                from dvd_refl
(b)                             _),
                              rw [mul_assoc,
x: ℕ                            nat.
h: x % 2 = 0                    mul_mod_right
⊢ x * x % 2 = 0                 ],
                            end
```

Figure 2: (a) A Lean theorem and a correct proof found by COPRA. (b) Proof state after the first tactic. (c) An incorrect proof generated by GPT-4.

Now we formalize our problem statement. Abstractly, let a *proof environment* consist of:

- A set of *states* $\mathcal{O}$. Here, a state is either a set $O = \{o_1, \ldots, o_k\}$ of obligations $o_i$, or of the form $(O, w)$, where $O$ is a set of obligations and $w$ is a textual *error message*. States of the latter form are *error states*, and the set of such states is denoted by *Err*.

- A set of *initial states*, each consisting of a single obligation $(g_{in}, h_{in})$ extracted from a user-provided theorem.

- A unique *goal state* QED is the empty obligation set.

- A finite set of *proof tactics*.

- A transition function $T(O, a)$, which determines the result of applying a tactic $a$ to a state $O$. If $a$ can be successfully applied at state $O$, then $T(O, a)$ is the new set of obligations resulting from the application. If $a$ is a "bad" tactic, then $T(O, a)$ is an error state $(O, w_e)$, where $w_e$ is some feedback that explains the error. Error states $(O, w)$ satisfy the property that $T((O, w), a) = (O, w)$ for all $a$.

- A set of *global contexts*, each of which is a natural-language string that describes the theorem to be proven and insights (Jiang et al., 2022b) about how to prove it. The theorem-proving agent can take a global context as an optional input and may use it to accelerate search.

Let us lift the transition function $T$ to tactic sequences by defining:

$$T(O, \alpha) = \begin{cases} T(O, a_1) & \text{if } n = 1 \\ T(T(O, \langle a_1, \ldots, a_{n-1} \rangle), a_n) & \text{otherwise.} \end{cases}$$

The theorem-proving problem is now defined as follows:

**Problem 1** (Theorem-proving). Given an initial state $O_{in}$ and an optional global context $\chi$, find a tactic sequence $\alpha$ (a *proof*) such that $T(O_{in}, \alpha) = \text{QED}$.

In practice, we aim for proofs to be as short as possible.

# 3   The COPRA Agent

The COPRA agent solves the theorem-proving problem via a GPT-4-directed depth-first search over tactic sequences. Figure 3 shows pseudocode for the agent. At any given point, the algorithm maintains a stack of environment states and a "failure dictionary" *Bad* that maps states to sets of tactics that are known to be "unproductive" at the state.

At each search step, it pushes the current state on the stack and retrieves external lemmas and definitions relevant to the state. After this, it serializes the stack, $Bad(O)$, the retrieved information, and the optional global context into a prompt and feeds it to the LLM. The LLM's output is parsed into a tactic and executed in the environment.

One outcome of the tactic could be that the agent arrives at `QED` or makes some progress in the proof. A second possibility is that the new state is an error. A third possibility is that the tactic is not incorrect but does not represent progress in the proof. We detect this scenario using a symbolic *progress check* as described below. The agent terminates successfully after reaching `QED` and rejects the new state in the second and third cases. Otherwise, it recursively continues the proof from the new state. After issuing a few queries to the LLM, the agent backtracks.

**Progress Checks.** Following Sanchez-Stern et al. (2020), we define a partial order $\sqsupseteq$ over states that indicates when a state is "at least as hard" as another. Formally, for states $O_1$ and $O_2$ with $O_1 \notin Err$ and $O_2 \notin Err$, $O_1 \sqsupseteq O_2$ iff

$$\forall o_i = (g_i, h_i) \in O_2. \qquad \exists o_k = (g_k, h_k) \in O_1. \ g_k = g_i \wedge (h_k \subseteq h_i).$$

Intuitively, $O_1 \sqsupseteq O_2$ if for every obligation in $O_2$, there is a stronger obligation in $O_1$. While comparing the difficulty of arbitrary proof states is not well-defined, our definition helps us eliminate some straightforward cases. Particularly, these cases include those proof states whose goals match exactly and one set of hypotheses contains the other.

As shown in Figure 3, the COPRA procedure uses the $\sqsupseteq$ relation to only take actions that lead to progress in the proof. Using our progress check, COPRA avoids cyclic tactic sequences that would cause nontermination.

**Prompt Serialization.** The routines PROMPTIFY and PARSETACTIC together constitute the *prompt serialization protocol* and are critical to the success of the policy. Now we elaborate on these procedures.

PROMPTIFY carefully places the different pieces of information relevant to the proof in the prompt. It also includes logic for trimming this information to fit the most relevant parts in the LLM's context window. Every prompt has two parts: the "system prompt" (see more details in Appendix A.3) and the "agent prompt."

The agent prompts are synthetically generated using a context-free grammar and contain information about the state stack (including the current proof state), the execution feedback for the previous action, and the set of actions we know to avoid at the current proof state.

```
COPRA(O, χ)
1   PUSH(st, O)
2   ρ ← RETRIEVE(O)
3   for j ← 1 to t
4       do p ← PROMPTIFY(st, Bad(O), ρ, χ)
5           a ∼ PARSETACTIC(LLM(p))
6           O' ← T(O, a)
7           if O' = QED
8               then terminate successfully
9               else if O' ∈ Err or
                        ∃O'' ∈ st. O' ⊒ O''
10                   then add a to Bad(O)
11                   else COPRA(O', χ)
12  POP(st)
```

Figure 3: The search procedure in COPRA. $T$ is the environment's transition function; $st$ is a stack, initialized to be empty. $Bad(O)$ is a set of tactics, initialized to $\varnothing$, that are known to be bad at $O$. LLM is an LLM, PROMPTIFY generates a prompt, PARSETACTIC parses the output of the LLM into a tactic (repeatedly querying the LLM in case there are formatting errors in its output), and RETRIEVE gathers relevant lemmas and definitions from an external source. The overall procedure is called with a state $O_{in}$ and an optional global context $\chi$.

The system prompt describes the rules of engagement for the LLM. It contains a grammar (distinct from the one for agent prompts) that we expect the LLMs to follow when it

| | Query #1 | Query #2 | Query #3 | Query #4 |
|---|---|---|---|---|
| Serialized Proof State | [GOALS] [GOAL] 1 x * x % 2 = 0 [HYPOTHESES] 1 [HYPOTHESIS] x : ℕ [HYPOTHESIS] h : x % 2 = 0 | | | [GOALS] [GOAL] 1 x % 2 * (x % 2) % 2 = 0 [HYPOTHESES] 1 [HYPOTHESIS] x : ℕ [HYPOTHESIS] h : x % 2 = 0 |
| Stack & Failure Dictionary | [LAST STEP] intro h, | [STEPS] [STEP] intro h, [LAST STEP] rw h, | [STEPS] [STEP] intro h, [INCORRECT STEPS] [STEP] rw h, [LAST STEP] apply nat.mul_mod_right, | [STEPS] [STEP] intro h, [LAST STEP] rw nat.mul_mod, |
| Interaction Result | [SUCCESS] [END] | [ERROR MESSAGE] Got error in 'rw h,': error: rewrite tactic failed, did not find instance of the pattern in the target expression x % 2 [END] | [ERROR MESSAGE] Got error in 'apply nat.mul_mod_right,': error: invalid apply tactic, failed to unify x * x % 2 = 0 with ?m_1 * ?m_2 % ?m_1 = 0 [END] | [SUCCESS] [END] |
| LLM Response | [RUN TACTIC] rw h, [END] | [RUN TACTIC] apply nat.mul_mod_right, [END] | [RUN TACTIC] rw nat.mul_mod, [END] | [RUN TACTIC] rw h, [END] |

Figure 4: A "conversation" between the LLM and the search algorithm in COPRA. Query #1, Query #2, . . . represent queries made as the proof search progresses. The column labeled Query #*i* depicts the prompt at time step *i* and the corresponding LLM response. The LLM response is parsed into a tactic and executed, with the error feedback incorporated into the next prompt.

proposes a course of action. The grammar carefully incorporates cases when the response is incomplete because of the LLM's token limits. We parse partial responses to extract the next tactic using the PARSETACTIC routine. PARSETACTIC also identifies any formatting errors in the LLM's responses, possibly communicating with the LLM multiple times until these errors are resolved.

Figure 4 illustrates a typical "conversation" between the LLM and COPRA's search algorithm via the prompt serialization protocol. Note, in particular, the prompt's use of the current goals, the stack, and error feedback for the last executed tactic.

## 4 Evaluation

**Benchmarks.** We primarily evaluate COPRA on the Lean proof environment using `miniF2F-test` (Zheng et al., 2021) benchmark. This benchmark comprises 244 formalizations of mathematics problems from (i) the MATH dataset (Hendrycks et al., 2021), (ii) high school mathematics competitions, (iii) hand-designed problems mirroring the difficulty of (ii). Broadly, these problems fall under three categories: number theory, algebra, and induction.

To evaluate COPRA's ability to work with multiple proof frameworks, we perform a secondary set of experiments on the Coq platform. Our benchmark here consists of a set of a theorems drawn from the CompCert compiler verification project (Leroy, 2009). The dataset was originally evaluated on the PROVERBOT9001 system (Sanchez-Stern et al., 2020). Due to budgetary constraints, our evaluation for CompCert uses 118 of the 501 theorems used in the original evaluation of PROVERBOT9001. For fairness, we include all 98 theorems proved by PROVERBOT9001 in our system. The remaining theorems are randomly sampled.

**Implementing COPRA.** Our implementation of COPRA is LLM-agnostic, and functions as long as the underlying language model is capable of producing responses which parse according to the output grammar. We instantiate the LLM to be GPT-4-turbo as a middle-ground between quality and affordability. Because of the substantial cost of GPT-4 queries, we cap the number of queries that COPRA can make at 60 for the majority of our experiments.

In both Lean and Coq, we instantiate the retrieval mechanism to be a BM25 search. For our Lean experiments, our retrieval database is `mathlib` (mathlib Community, 2020), a mathematics library developed by the Lean community. Unlike `miniF2F`, our CompCert-based evaluation set is accompanied by a training set. Since COPRA relies on a black-box LLM, we do not perform any training with the train set, but do use it as the retrieval database for our Coq experiments.

Inspired by DSP (Jiang et al., 2022b), we measure the efficacy of including the natural-language (informal) proof of the theorem at each search step in our Learn experiments. We use the informal theorem statements in `miniF2F-test` to generate informal proofs of theorems using few-shot prompting of the underlying LLM. The prompt has several examples, fixed over all invocations, of informal proof generation given the informal theorem statement. These informal proofs are included as part of the global context. Unlike `miniF2F`, which consists of competition problems originally specified in natural language, our CompCert evaluation set comes from software verification tasks that lack informal specifications. Hence, we do not run experiments with informal statements and proofs for Coq.

Initially, we run COPRA without access to the retrieval database. By design, COPRA may backtrack out of its proof search before the cap of 60 queries has been met. With the remaining attempts on problems yet to be solved, COPRA is restarted to operate with retrieved information, and then additionally with informal proofs. This implementation affords lesser expenses, as appending retrieved lemmas and informal proofs yields longer prompts, which incur higher costs. More details are discussed in Appendix A.1.1.

**Baselines.** We perform a comparison with the few-shot invocation of GPT-4 in both the `miniF2F-test` and CompCert domains. The "system prompt" in the few-shot approach differs from that of COPRA, containing instructions to generate the entire proof in one response, rather than going step-by-step. For both COPRA and the few-shot baseline, the prompt contains several proof examples which clarify the required format of the LLM response. The proof examples, fixed across all test cases, contain samples broadly in the same categories as, but distinct from, those problems in the evaluation set.

For our fine-tuned baselines, a challenge is that, to our knowledge, no existing theorem-proving system targets *both* Lean and Coq. Furthermore, all open-source theorem proving systems have targeted a single proof environment. As a result, we had to choose different baselines for the Lean (`miniF2F`) and Coq (CompCert) domains.

Our fine-tuned baseline for `miniF2F-test` is REPROVER, a state-of-the-art retrieval-augmented open-source system that is part of the LeanDojo project (Yang et al., 2023). We also compare against LLEMMA-7b and LLEMMA-34b (Azerbayev et al., 2023). In the CompCert task, we compare with PROVERBOT9001 (Sanchez-Stern et al., 2020), which, while not LLM-based, is the best public model for Coq.

**Ablations.** We consider ablations of COPRA which use GPT-3.5 and CodeLlama (Roziere et al., 2023) as the underlying model. We also measure the effectiveness of backtracking in COPRA's proof search. Additionally, we ablate for retrieved information and informal proofs as part of the global context during search.

**Metric.** The standard metric for evaluating theorem-provers is *pass@k* (Lample et al., 2022). In this metric, a prover is given a budget of *k proof attempts*; the method is considered successful if one of these attempts leads to success. We consider a refinement of this metric called *pass@k-with-n-queries*. Here, we measure the number of correct proofs that a prover can generate given *k* attempts, each with a budget of *n* queries from the LLM or neural model. We enforce that a single query includes exactly one action (a sequence of tactics) to be used in the search. We want this metric to be correlated with the number of correct proofs that the prover produces within a specified wall-clock time budget; however, the cost of an inference query on an LLM is proportional to the number of responses generated per query. To maintain the correlation between the number of inference queries and wall-clock time, we restrict each inference on the LLM to a single response (see Appendix A.1.2 for more details).

| Evaluation on `miniF2F-test` | | |
|---|---|---|
| **Approach** | *pass@k -with-n-queries* $k \times n$ **(Timeout)** | % **proved** |
| Few-Shot (CodeLlama) | $1 \times 1$ (600) | 0.0% |
| Few-Shot (GPT 3.5) | $1 \times 1$ (600) | 2.8% |
| COPRA (CodeLlama) | $1 \times 500$ (600) | 5.73% |
| COPRA (GPT-3.5) | $1 \times 60$ (600) | 9.02% |
| Few-Shot | $1 \times 1$ (600) | 13.52% |
| Few-Shot ($T = 0.7$) | $60^{\ddagger} \times 1$ (600) | 15.98% |
| REPROVER (- Retrieval) | $1 \times 1076$ (600) | 22.13% |
| COPRA (- Backtracking) | $1 \times 60$ (600) | 24.59% |
| REPROVER | $1 \times 3751$ (600) | 25.00% |
| LLEMMA-34b | $1 \times 3200$ (600) | 25.82% |
| LLEMMA-7b | $1 \times 3200$ (600) | 26.23% |
| **COPRA** | $1 \times 60$ (600) | **26.63%** |
| **COPRA (+ Retrieval)** | $1 \times 60$ (600) | **29.09%** |
| **COPRA (+ Retrieval + Informal)** | $1 \times 60$ (600) | **29.92%** |
| **COPRA (+ Retrieval + Informal)** | $1 \times 100$ (1200) | **30.74%** |

Table 1: Aggregate statistics for COPRA and the baselines on `miniF2F` dataset. Note the various ablations of COPRA with and without retrieval or informal proofs. The timeout is a wall-clock time budget in seconds allocated across all attempts. Unless otherwise specified, (i) COPRA uses GPT-4 as the LLM (ii) the temperature ($T$) is set to 0.

**COPRA vs. Few-Shot LLM Queries.** Statistics for the two approaches, as well as a comparison with the few-shot GPT-3.5 and GPT-4 baselines, appear in Table 1. As we see, CO-PRA offers a significant advantage over the few-shot approaches. For example, COPRA solves more than twice as many problems as the few-shot GPT-4 baseline, which indicates that the state information positively assists the search. Furthermore, we find that running the few-shot baseline with $60^{\ddagger}$ attempts (provided a nonzero temperature), exhibits considerably worse performance compared to COPRA. This indicates that COPRA makes more efficient use of queries to GPT-4 than the baseline, even when queries return a single tactic (as part of a stateful search), as opposed to a full proof. We include further details in Appendix A.1.1.

COPRA is capable of improving the performance of GPT-3.5, CodeLlama over its few-shot counterpart. We note that the use of GPT-4 seems essential, as weaker models like GPT-3.5 or CodeLlama have a reduced ability to generate responses following the specified output format.

**Comparison with Finetuned Approaches on `miniF2F`.** Figure 5 shows our comparison results for the `miniF2F` domain. With a purely in-context learning approach, CO-PRA outperforms REPROVER, proving within just 60 queries theorems that REPROVER could not solve after a thousand queries. This is remarkable given that REPROVER was finetuned on a curated proof-step dataset derived from Lean's `mathlib` library. For fairness, we ran REPROVER multiple times with 16, 32, and 64 (default) as the maximum number of queries per proof-step. We obtained the highest success rates for REPROVER with 64 queries per proof-step. We find that COPRA without retrieval outperforms REPROVER.

Our *pass@1* performance surpasses those of the best open-source approaches for `miniF2F-test` on Lean. COPRA proves 29.09% of theorems in `miniF2F-test` theorems, which exceeds that of LLEMMA-7b & LLEMMA-34b, (Azerbayev et al., 2023) and REPROVER (Yang et al., 2023). It is important to note that these methods involve training on curated proof data, while COPRA relies only on in-context learning.

We also establish a correlation between the number of *queries* needed for a proof and wall-clock time in Figure 6 and Table 3 (more details are discussed in Appendix A.1.3). Although the average time per query is higher for COPRA, COPRA still finds proofs almost 3x faster than REPROVER. This can explained by the fact that our search is more effective as it uses 16x fewer queries than REPROVER (see Table 2). The average time per query includes the time taken to execute the generated proof step by the interactive theorem prover. Hence, a more

---

$^{\ddagger}$To ensure fairness we matched the number of attempts for Few-Shot (GPT-4) baseline with the number of queries COPRA took to pass or fail for each theorem.

effective model can be slow in generating responses while still proving faster compared to a model with quicker inference, by avoiding the waste of execution time on bad proof-steps.

**Qualitative Analysis of Proofs.** We performed an analysis of the different categories of `miniF2F` problems solved by COPRA and REPROVER. Figure 8 and Figure 9 (in Appendix A.2) show that COPRA proves more theorems in most of the categories and takes fewer steps consistently across all categories of problems in `miniF2F` as compared to REPROVER. Additionally, we find that certain kinds of problems, for example, International Mathematics Olympiad (IMO) problems and theorems that require induction, are difficult for both approaches.

From our qualitative analysis, there are certain kinds of problems where our language-agent approach seems especially helpful. For instance, Figure 7 shows a problem in the 'numbertheory' category that REPROVER could not solve. More examples of interesting proofs that COPRA found appear in the Appendix A.2 in Figure 10.

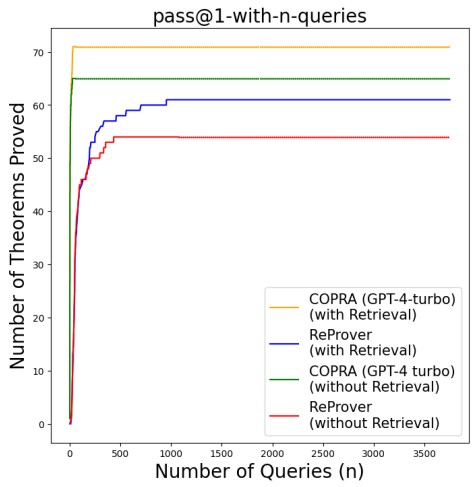

Figure 5: COPRA vs. REPROVER on the `miniF2F` benchmark.

**Effectiveness of Backtracking, Informal Proofs, and Retrieval.** We show the ablation for the backtracking feature of COPRA in Table 1. We find that backtracking is useful when proofs are longer or more complex, as COPRA is more likely to make mistakes which require amending. We include additional examples in Figure 12 (in Appendix A.2).

From Table 1, we see that retrieval helps in proving more problems. Retrieval reduces the tendency of the model to hallucinate the lemma names and provides the model with existing lemma definitions it otherwise may not have known. We include a proof generated by COPRA that uses a retrieved lemma in Figure 7.

We experiment with adding model-generated informal proofs as global context in COPRA's proof search. As evidenced in Table 1, COPRA is able to outperform PACT (Han et al., 2021) and the state-of-the-art (29.6%) expert iteration method (Polu et al., 2022) in a *pass*@1 search through the incorporation of informal proofs. Furthermore, increasing the maximum query count to 100 enables a further increase in COPRA's performance to 30.74%. An example of a proof found when incorporating informal proofs is shown in Figure 15 (see Appendix A.5).

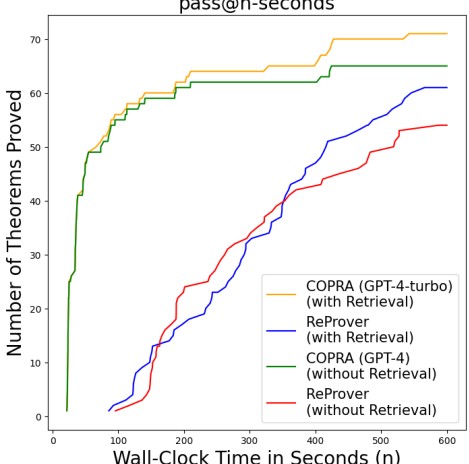

Figure 6: COPRA vs. REPROVER on the `miniF2F` benchmark on the *pass@n-seconds* metric.

**Test-Set Memorization Concerns.** Given that the pretraining corpus of GPT-4 is not publicly available, it is imperative to assess for the possibility of test-set leakage in our experiments. We perform an analysis of those proofs checked into the open-source `miniF2F` repository compared to those proofs COPRA discovers.

We categorize those proofs available in `miniF2F` according to the number of tactics required and the complexity of the tactic arguments. These results can be found in detail in Appendix A.1.4. As seen in Table 4, we find that COPRA reproduces none of the "long" proofs

in `miniF2F-test` repository. Setting aside those cases where the proof consists of a single tactic, approximately 91.93% of the proofs generated by COPRA either bear no overlap to proofs in `miniF2F`, or no proof has been checked in. We provide examples of proofs generated by our approach compared to those included in `miniF2F` in Figure 11.

```
theorem mathd_numbertheory_100
(n : ℕ)
(h₀ : 0 < n)
(h₁ : nat.gcd n 40 = 10)
(h₂ : nat.lcm n 40 = 280) :
n = 70 :=
begin
    have h₃ : n * 40 = 10 * 280 := by
    rw ←[nat.gcd_mul_lcm n 40, h₁, h₂],
    exact (nat.eq_of_mul_eq_mul_right (
    by norm_num : 0 < 40) h₃),
end
```

Figure 7: A theorem that requires retrieval. CO-PRA used BM25 to find the lemma "gcd_mul_lcm : $(n\ m : \mathbb{N}+) : (\text{gcd}\ n\ m) * (\text{lcm}\ n\ m) = n * m$" which led to the proof while REPROVER failed to prove this theorem. It is important to note that just retrieving the correct lemma is not sufficient, but knowing how to use it correctly is equally important. The Figure 18 shows how COPRA utilizes the capabilities of LLM to correctly use the retrieved lemma in the proof.

**Coq Experiments** COPRA can prove a significant portion of the theorems in our Coq evaluation set. As shown in Figure 16 (in Appendix A.5), COPRA slightly outperforms PROVERBOT9001 when both methods are afforded the same number of queries. COPRA with retrieval is capable of proving 57 of 118 theorems within our cap of 60 queries. Furthermore, COPRA exceeds the few-shot baselines utilizing GPT-4 and GPT-3.5, which could prove 36 and 10 theorems, respectively, on our CompCert-based evaluation set. Some example Coq proofs generated by COPRA are shown in Appendix A.5 in Figure 17.

## 5 Related Work

**Neural Theorem-Proving.** There is a sizeable literature on search-based theorem-proving techniques based on supervised learning. Neural models are trained to predict a proof step given a context and the proof state, then employed to guide a search algorithm (e.g. best-first or depth-limited search) to synthesize the complete proof.

Early methods of this sort (Yang & Deng, 2019; Sanchez-Stern et al., 2020; Huang et al., 2019) used small-scale neural networks as proof step predictors. Subsequent methods, starting with GPT-$f$ (Polu & Sutskever, 2020), have used language models trained on proof data.

PACT (Han et al., 2021) enhanced the training of such models with a set of self-supervised auxiliary tasks. Lample et al. (2022) introduced HyperTree Proof Search, which uses a language model trained on proofs to guide an online MCTS-inspired, search algorithm.

Among results from the very recent past, REPROVER trains a retrieval-augmented transformer for proof generation in Lean. LLEMMA performs continued pretraining of the CodeLlama 7B & 34B on a math-focused corpus. Baldur (First et al., 2023) generates the whole proof in one-shot using an LLM and then performs a single *repair* step by passing error feedback through an LLM finetuned on (incorrect proof, error message, correct proof) tuples. AlphaGeometry (Trinh et al., 2024) integrates a transformer model trained on synthetic geometry data with a symbolic deduction engine to prove olympiad geometry problems. In contrast to these approaches, COPRA is entirely based on in-context learning.

We evaluated COPRA in the Lean and Coq environments. However, significant attention has been applied to theorem proving with LLMs in the interactive theorem prover Isabelle (Paulson, 1994). Theorem-proving systems for Lean and Isabelle are not directly comparable due to the substantial differences in automation provided by each language. Isabelle is equipped with Sledgehammer (Paulson & Blanchette, 2015), a powerful automated reasoning tool that calls external automated theorem provers such as E (Schulz, 2002) and Z3 (De Moura & Bjørner, 2008) to prove goals. Thor (Jiang et al., 2022a) augmented the PISA dataset (Jiang et al., 2021) to include successful Sledgehammer invocations, and trained a language model to additionally predict hammer applications. Integrating these ideas with the COPRA approach is an interesting subject of future work.

The idea of using informal hints to guide proofs was first developed in DSP (Jiang et al., 2022b), which used an LLM to translate informal proofs to formal sketches that were then completed with Isabelle's automated reasoning tactics. Zhao et al. (2023) improved on DSP by rewriting the informal proofs to exhibit a more formal structure and employs a diffusion model to predict the optimal ordering of the few-shot examples. LEGOProver (Wang et al., 2023b) augmented DSP with a skill library that grows throughout proof search. Lyra (Zheng et al., 2023) iterated on DSP by utilizing error feedback several times to modify the formal sketch, using automated reasoning tools to amend incorrect proofs of intermediate hypotheses. DSP, LEGOProver, and Lyra heavily rely on Isabelle's hammer capabilities (which are not present in other ITPs like Lean) and informal proofs for formal proof generation (see Appendix A.1.5 for details). Moreover, the requirement for informal proofs in these approaches prohibits their application in the software verification domain, where the notion of informal proof is not well-defined. However, we propose a domain-agnostic stateful search approach within an in-context learning agent which iteratively uses execution feedback from the ITP with optional use of external signals like retrieval and informal proofs.

**In-Context Learning Agents.** Several distinct in-context learning agent architectures have been proposed recently (Significant-Gravitas, 2023; Yao et al., 2022; Shinn et al., 2023; Wang et al., 2023a). These models combine an LLM's capability to use tools Schick et al. (2023), decompose a task into subtasks (Wei et al., 2022; Yao et al., 2023), and self-reflect (Shinn et al., 2023). However, COPRA is the first in-context learning agent for theorem-proving.

## 6 Conclusion

We have presented COPRA, the first in-context-learning approach to theorem-proving in frameworks like Lean and Coq. The approach departs from prior LLM-based theorem-proving techniques by performing a *history-dependent* backtracking search by utilizing in-context learning, the use of execution feedback from the underlying proof environment, and retrieval from an external database. We have empirically demonstrated that COPRA significantly outperforms few-shot LLM invocations at proof generation and also compares favorably to finetuned approaches.

As for future work, we gave GPT-4 a budget of at most 60 queries per problem for cost reasons. Whether the learning dynamics would drastically change with a much larger inference budget remains to be seen. Also, it is unclear whether a GPT-4-scale model is truly essential for our task. We have shown that the cheaper GPT-3.5 agent is not competitive against our GPT-4 agent; however, it is possible that a Llama-scale model that is explicitly finetuned on interactions between the model and the environment would have done better. While data on such interactions is not readily available, a fascinating possibility is to generate such data *synthetically* using the search mechanism of COPRA.

## 7 Reproducibility Statement

We are releasing all the code needed to run COPRA as supplementary material. The code contains all "system prompts" described in Section Appendix A.3 and Appendix A.4, along with any other relevant data needed to run COPRA. However, to use our code, one must use their own OpenAI API keys. An issue with reproducibility in our setting is that the specific models served via the GPT-4 and GPT-3.5 APIs may change over time. In our experiments, we set the "temperature" parameter to zero to ensure the LLM outputs are as deterministic as possible.

**Funding Acknowledgements.** This work was partially supported by NSF awards CCF-1918651, CCF-2403211, and CCF-2212559, and a gift from the Ashar Aziz Foundation.

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

# A   Appendix

## A.1   Evaluation Details

### A.1.1   COPRA Implementation Setup Details

We introduce a common proof environment for COPRA, which can also be used by any other approach for theorem-proving. The proof environment has a uniform interface that makes COPRA work seamlessly for both Lean and Coq. It also supports the use of retrieval from external lemma repositories and the use of informal proofs when available (as in the case of `miniF2F`). In the future, we plan to extend COPRA to support more proof languages.

COPRA provides support for various LLMs other than the GPT-series, including open-sourced LLMs like Llama 2 (Touvron et al., 2023) and Code Llama (Roziere et al., 2023). For most of our experiments, all the theorems are searched within a timeout of 600 seconds (10 minutes) and with a maximum of 60 LLM inference calls (whichever exhausts first). To make it comparable across various LLMs, only one response is generated per inference. All these responses are generated with the *temperature* set to zero, which ensures that the responses generated are more deterministic, focused, and comparable. In one of our ablations with few-shot GPT-4, we use set the *temperature* to 0.7 and run more than 1 attempt. The number of attempts for few-shot GPT-4 matches the number of queries COPRA takes to either successfully prove or fail to prove a certain theorem. This comparison is not completely fair for COPRA because individual queries focus on a single proof step and cannot see the original goal, yet measures a useful experiment which indicates the benefits of COPRA's stateful search under a fixed budget of API calls to GPT-4.

We use GPT-3.5, GPT-4 (OpenAI, 2023b), and CodeLLama (Roziere et al., 2023) to test the capabilities of COPRA. We find that it is best to use COPRA's different capabilities in an ensemble to enhance its performance while also minimizing cost. Therefore, we first try to find the proof without retrieval, then with retrieval, and finally with informal proofs (if available) only when we fail with retrieval. The informal proofs are used more like informal sketches for the formal proof and are generated in a separate few-shot invocation of the LLM. Thereafter, COPRA simply adds the informal proofs to its prompt which guides the search. To ensure fairness in comparison, we make sure that the number of guidance steps is capped at 60 and the 10-minute timeout is spread across all these ensemble executions. The use of a failure dictionary (see Section 3) enables fast failure (see Table 3) which helps in using more strategies along with COPRA within the 60 queries cap and 10-minute timeout. From Table 2, it is clear that despite the significant overlap between the different ablations, the ensemble covers more cases than just using one strategy with COPRA. While extra lemmas are often useful, the addition of extra information from retrieval may also be misleading because the retriever can find lemmas that are not relevant to proving the goal, so the best performance is acquired by using the different capabilities in the manner we describe. Similarly, the use of informal proofs as a sketch for formal proofs can potentially increase the number of steps needed in a formal proof, decreasing the search efficacy. For example, an informal proof can suggest the use of multiple rewrites for performing some algebraic manipulation which can be easily handled with powerful tactics like `linarith` in Lean. Table 5 shows the number of tokens used in our prompts for different datasets.

### A.1.2   Metric: *pass@k-with-n-queries*

We consider the metric *pass@k-with-n-queries* to assess the speed of the proposed approach and the effectiveness of the LLM or neural network to guide the proof search. It is a reasonable metric because it does a more even-handed trade-off in accounting for the time taken to complete a proof and at the same time ignores very low-level hardware details.

Different approaches need a different amount of guidance from a neural model to find the right proof. For example, approaches like Baldur (First et al., 2023), DSP (Jiang et al., 2022b), etc., generate the whole proof all at once. On the other hand, GPT-*f* (Polu & Sutskever, 2020), PACT (Han et al., 2021), REPROVER (Yang et al., 2023), Proverbot (Sanchez-Stern et al., 2020), and COPRA generate proofs in a step-by-step fashion. We argue that *pass@k-with-n-*

| Evaluation on `miniF2F-test` | | | |
|---|---|---|---|
| Approach | Avg. Query in Total | Avg. Query on Failure | Avg. Query on Pass |
| REPROVER (- Retrieval) | 350.7 | 427.24 | 81.6 |
| REPROVER | 1015.32 | 1312.89 | 122.62 |
| COPRA | 21.73 | 28.23 | 3.83 |
| COPRA (+ Retrieval) | 37.88 | 50.90 | 7.38 |

Table 2: Aggregate effectiveness statistics for COPRA and the baselines on `miniF2F` dataset. We can see the various ablations of COPRA with and without retrieval.

| Approach | Avg. Time In Seconds | | | | | |
|---|---|---|---|---|---|---|
| | Per Proof | | | Per Query | | |
| | On Pass | On Fail | All | On Pass | On Fail | All |
| ReProver (on CPU - retrieval) | 279.19 | 618.97 | 543.78 | 3.42 | 1.45 | 1.55 |
| ReProver (on GPU - retrieval) | 267.94 | 601.35 | 520.74 | 2.06 | 0.44 | 0.48 |
| ReProver (on GPU + retrieval) | 301.20 | 605.29 | 529.27 | 2.46 | 0.46 | 0.52 |
| COPRA (GPT-3.5) | 39.13 | 134.26 | 122.21 | 15.97 | 9.43 | 9.53 |
| **COPRA (GPT-4)** | **67.45** | 370.71 | 289.92 | 17.61 | 13.09 | 13.34 |
| **COPRA (GPT-4 + retrieval)** | **117.85** | 678.51 | 510.78 | 15.97 | 13.33 | 13.48 |

Table 3: Average time taken by our approach (COPRA) and REPROVER on `miniF2F` dataset. We split the values according to the success of the proof search on that problem. We also report values per query.

*queries* is a fairer metric to compare these different types of approaches because it correlates with the effectiveness of the proof-finding algorithm in an implementation-agnostic way. Since the exact time might not always be a good reflection of the effectiveness because of hardware differences, network throttling, etc., it makes sense to not compare directly on metrics like pass@k-minutes or pass@k-seconds. Not only these metrics will be brittle and very sensitive to the size, hardware, and other implementation details of the model, but not every search implementation will be based on a timeout. For example, PROVERBOT9001 does not use timeout-based search (and hence we don't compare on the basis of time with PROVERBOT9001).

### A.1.3 *pass@k-with-n-queries* versus wall-clock time

We show that *pass@k-with-n-queries*, correlates well with wall-clock time for finding proofs by using the metric *pass@k-seconds*. *pass@k-seconds* measures the number of proofs that an approach can find in less than *k* seconds. The plot in Figure 6 shows that *pass@k-seconds* follows the same trend as *pass@k-with-n-queries* as shown in Figure 5.

We can use the comparison of COPRA with REPROVER (Yang et al., 2023) on the miniF2F dataset to explain the correlation between finding proofs fast and *pass@k-with-n-queries*. From Table 3, we know that on average the time taken per guidance (which includes time taken to execute the proof steps on ITP as well) is around 0.52 seconds for REPROVER and 13.48 seconds for COPRA. Given that REPROVER's guidance LLM is small, we can assume that REPROVER does not take any time (zero time) to query its LLM and spends most of the 0.52 seconds running the proof steps on ITP. Now, we can reasonably estimate GPT-4 average response time to be approximately 13 seconds from Table 3. However, we see that the number of guidance used by REPROVER (from Table 2) is about 16x higher on success. Interestingly, this also shows up in the wall clock time, which is around 3x higher for REPROVER on success, so there is a tradeoff between the two, but the number of queries dominates when the guidance model is of low quality. Hence, given a high-quality guidance model, we can empirically argue that asymptotically the search will converge to a proof faster (assuming a proof is achievable with the guidance model).

### A.1.4 Data Leakage in GPT-4

A key risk with closed-source pretrained models like GPT-4 is data leakage, i.e., overlaps between the evaluation set and the pretraining set. Naturally, we cannot be *certain* that COPRA does not benefit from such leakage. However, there are several reasons to believe that data leakage is not a significant contributor to our results.

First, we note that COPRA significantly outperforms few-shot invocations of GPT-4. If the results on COPRA were significantly tainted by data leakage, we would have expected better performance from few-shot GPT-4.

Second, it is highly unlikely that GPT-4 has been trained on proof-state and tactic pair generated by hooking up the Lean ITP. Not all the formal proofs of the miniF2F test dataset are available online (only 80 proofs are available in Lean). Furthermore, if one were to manually annotate (proof state, tactic) pairs, one would need ground truth tactics to annotate with, the majority of which do not appear in `miniF2F-test`. Given that GPT-4 is a general-purpose LLM, it is highly unlikely that while training GPT-4 the `miniF2F` dataset was first manually annotated, and then proof-state and tactic pair information was collected by hooking up the Lean ITP.

Also, in our agent interactions, we limit ourselves only to the goal at that point. There is no mention of the original theorem anywhere (except for the very first proof-state), so the chances that GPT-4 can correlate any intermediate state with the original theorem are slim, unless it has learned a model of Lean's kernel, which is highly unlikely. It is also improbable that GPT-4 has seen the proof-state in the same format that we use, let alone using the execution feedback which has not been used in any known previous works for Lean.

One could hypothesize that some of the few-shot GPT-4 proofs might be influenced by potential training on the `miniF2F` dataset. However, this does not seem to be true because we see that most of the proofs we generated were either not mentioned in the `miniF2F` test dataset or completely different from the manually written proofs in the `miniF2F` test dataset (including the first step mismatch). Table 4 shows the detailed analysis of proofs generated by COPRA and the proofs mentioned in `miniF2F` test dataset for Lean. From the Table 4, it is clear that most of the proofs generated by COPRA are different from the proofs mentioned in the `miniF2F`. The ones that are exactly the same are simple single-tactic proofs that just use exactly one of the `linarith`, `nlinarith`, or `norm_num` tactics without any arguments. If we set aside these straightforward simple cases, then about 91.93% of the proofs generated by COPRA are either different from the proofs mentioned in the `miniF2F` or do not have a proof mentioned in the `miniF2F` dataset. Out of all proofs generated by COPRA about 25.33% proofs are for theorems that have no proofs mentioned in the `miniF2F` test dataset as compared to 22.95% for REPROVER. Some of the proofs generated by our approach as compared to proofs mentioned in the `miniF2F` test dataset are shown in Figure 11.

Finally, the ability of agent interactions to enhance the basic LLM approach seems to transcend OpenAI's LLMs. We ran COPRA on the recently released CodeLlama. From Table 2, COPRA improved CodeLlama's capabilities to prove theorems by about 5% on `miniF2F` dataset. This indicates that the in-context learning capabilities that we build are transferable and LLM-agnostic.

### A.1.5 Comparison with methods using Isabelle and informal proofs

An interactive theorem prover (ITP) is a software tool to assist with the development of formal proofs by human-machine collaboration. This involves a sort of interactive proof editor, or other interfaces, with which a human can guide the search for proofs. A formal proof written in an interactive theorem prover can be verified automatically by computers, whereas an informal proof is written in natural language can only be verified by a human. Generally, formal proofs are much more rigorous and pedantic than informal proofs. So informal proof can be loosely considered as a proof sketch based on which one can write rigorous machine-checkable formal proofs.

Note that the accuracy numbers of DSP-like approaches (Jiang et al., 2022b; Zhao et al., 2023; Zheng et al., 2021) are not directly comparable to ours because they use a differ-

| | | Proofs found in `miniF2F-test` | | | | Proofs NOT in `miniF2F` | Total |
|---|---|---|---|---|---|---|---|
| | | Single-Tactic Simple Proofs | | | Two-Tactic Proofs | Longer OR Complex Proofs | Total | | |
| Tactics Used —— Proof Count | `linarith` | `norm_num` | `nlinarith` | two tactics | > 2 tactics OR 1 tactic multi-args | | **sorry** | |
| Proof Count | 11 | 12 | 2 | 16 | 39 | 80 | 164 | 244 |
| Exact Match COPRA Count | 7 | 10 | 0 | 5 | 0 | 22 | 0 | 22 |
| 1$^{st}$ Tactic Match COPRA Count | 7 | 10 | 0 | 8 | 4 | 29 | 0 | 29 |
| Distinct COPRA Count | 4 | 2 | 2 | 9 | 17 | 34 | 19 | 53 / 75 **70.67%** |
| Distinct COPRA Count ex Single-Tactic | - | - | - | 9 | 17 | 34 | 19 | 53 / 58 **91.37%** |
| All COPRA Count | 11 | 12 | 2 | 14 | 17 | 56 | 19 | 75 |

Table 4: Analysis of proof generated by COPRA (GPT-4 + Retrieval + Informal) on `miniF2F` test dataset for Lean. See Table 1 for details about various ablations.

ent proof language. These approaches use Isabelle, which, unlike Lean, allows the use of powerful automatic reasoning tools like Sledgehammer. Methods following the DSP pipeline use informal proofs for formal proof synthesis. While this strategy works well on mathematics-competition benchmarks like `miniF2F`, it is less applicable to domains such as software verification, where there is often no informal specification, as well as domains that use customized, domain-specific formalizations. Furthermore, having access to informal proofs (human-written or LLM-generated) shifts the problem of synthesizing the formal proof towards an autoformalization problem, as the LLM is likely to have seen correct natural language proofs of `miniF2F` problems in its training. Additionally, unlike DSP-like approaches which tend to use *pass@100* or *pass@200*, we only use *pass@1*.

## A.2 Example Proofs Generated for `miniF2F`

Figure 10 shows some other interesting proofs generated by our approach on `miniF2F` dataset. Figure 8 and Figure 9 shows the breakdown of theorems proved in various categories by COPRA versus REPROVER. Figure 10 shows some interesting `miniF2F` proofs as done by COPRA. Figure 11 shows the comparison of proofs mentioned in the `miniF2F` repository versus the proofs discovered by COPRA. Figure 12 shows the proofs which were only possible because of our backtracking feature.

## A.3 System Prompts For `miniF2F`

Parts of the 'system prompt' used by COPRA for theorem proving in Lean are shown in Fig. 13.

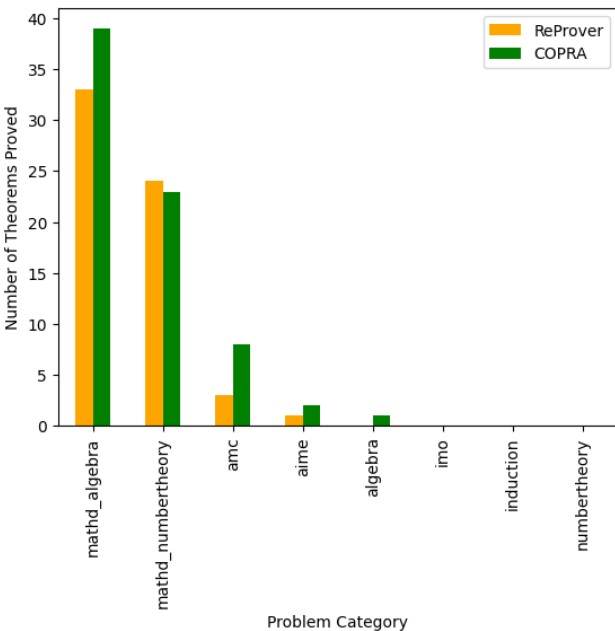

Figure 8: Breakdown of theorems proved in various categories by COPRA (GPT-4 + retrieval + informal proof): Problems solved in different categories in `miniF2F` test dataset. Notice that 'IMO' and 'Induction' problems are hard for both approaches. COPRA does more problems than REPROVER in most of the categories.

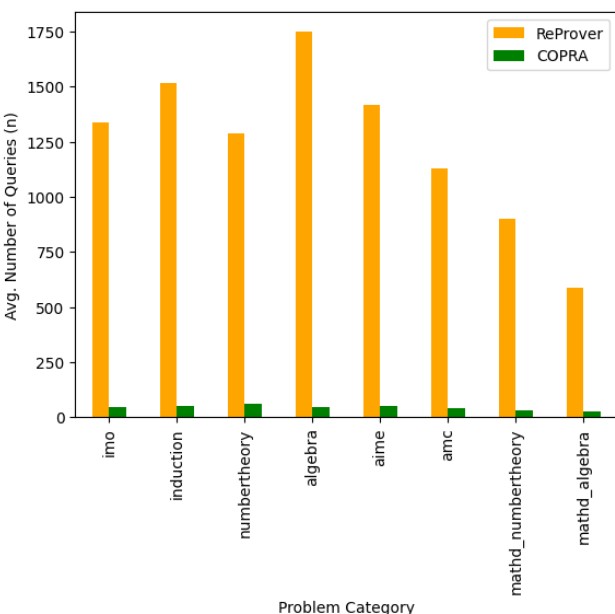

Figure 9: Breakdown of theorems proved in various categories by COPRA (GPT-4 + retrieval + informal proof): Number of queries needed in different categories in `miniF2F` test dataset. Notice that COPRA takes very less queries to solve the problems in each category.

(a)

```
theorem
    mathd_numbertheory_100
(n : ℕ)
(h₀ : 0 < n)
(h₁ : nat.gcd n 40 = 10)
(h₂ : nat.lcm n 40 = 280) :

n = 70 :=
begin
    have h₃ : n * 40 = 10 *
     280 := by rw ←[nat.
    gcd_mul_lcm n 40, h₁,
    h₂],
exact (nat.
    eq_of_mul_eq_mul_right
     (by norm_num : 0 <
    40) h₃),
end
```

(b)

```
theorem
    mathd_numbertheory_447
     :
∑ k in finset.filter (˘ x,
    |3x) (finset.Icc 1
    49), (k % 10) = 78 :=
begin
    norm_num [finset.
    sum_filter, finset.Icc
    ],
    ring,
end
```

(c)

```
theorem mathd_algebra_158
(a : ℕ)
(h₀ : even a)
(h₁ : ↑∑ k in finset.
    range 8, (2 * k + 1) -
     ↑∑ k in finset.
    range 5, (a + 2 * k)
    = (4:ℤ)) :
a = 8 :=
begin
    simp only [finset.
    sum_range_succ,
    finset.sum_range_zero,
     add_zero, finset.
    sum_const, finset.
    card_range,
    nsmul_eq_mul, finset.
    sum_add_distrib,
    finset.mul_sum,
    finset.sum_mul] at h₁,

    norm_num at h₁,
    linarith,
end
```

(d)

```
theorem
    mathd_numbertheory_341
(a b c : ℕ)
(h₀ : a ≤ 9 ∧ b ≤ 9 ∧ c
     ≤ 9)
(h₁ : nat.digits 10
     ((5^100) % 1000) = [c,
    b,a]) :
a + b + c = 13 :=
begin
    norm_num at h₁,
    cases h₁,
linarith,
end
```

(e)

```
theorem amc12b_2002_p7
(a b c : ℕ)
(h₀ : 0 < a ∧ 0 < b ∧ 0 < c)
(h₁ : b = a + 1)
(h₂ : c = b + 1)
(h₃ : a * b * c = 8 * (a + b
     + c)) :
a^2 + (b^2 + c^2) = 77 :=
begin
    rw [h₁, h₂] at h₃,
    ring_nf at h₃,
    rw [h₁, h₂],
    ring_nf,
    simp [h₁, h₂] at *,
    ring_nf at *,
    nlinarith,
end
```

Figure 10: Some other interesting proofs generated for `miniF2F` by COPRA. The length of the proofs generated shows that interaction with the environment helps in fixing the errors encountered while writing long proofs. These long sequences of rewrites are not easy to synthesize without knowing the exact execution feedback from the environment which often contains the hint to fix the rewrites.

(a.1)

```
theorem mathd_numbertheory_3 :
    (∑ x in finset.range 10, ((x + 1)^2)) %
        10 = 5 :=
begin
  dec_trivial!,
end
```

(b.1)

```
theorem mathd_algebra_209
  (σ : equiv ℝ ℝ)
  (h₀ : σ.2 2 = 10)
  (h₁ : σ.2 10 = 1)
  (h₂ : σ.2 1 = 2) :
  σ.1 (σ.1 10) = 1 :=
begin
  rw ←[ h₀, ← h₂],
  simp,
end
```

(c.1)

```
theorem mathd_numbertheory_229 :
  (5^30) % 7 = 1 :=
begin
  have five_to_thirty_is_one :
    (5^30 : zmod 7) = 1 :=
  begin
    have five_to_the_six_is_one : (5^6 :
    zmod 7) = 1, by dec_trivial,
    have break_power : (5^30 : zmod 7) =
    (5^6)^5, by norm_num,
    rw break_power,
    rw five_to_the_six_is_one,
    norm_num,
  end,
  change 5^30 ≡ 1 [MOD 7],
  rw ←zmod.eq_iff_modeq_nat,
  exact_mod_cast five_to_thirty_is_one,
end
```

(a.2)

```
theorem mathd_numbertheory_3 :
(∑ x in finset.range 10, ((x + 1)^2)) %
    10 = 5 :=
begin
    simp only [finset.sum_range_succ,
      finset.sum_range_zero, add_zero,
      pow_two],
    norm_num,
end
```

(b.2)

```
theorem mathd_algebra_209
  (σ: equiv ℝ ℝ)
  (h₀ : σ.2 2 = 10)
  (h₁ : σ.2 10 = 1)
  (h₂ : σ.2 1 = 2) :
  σ.1 (σ.1 10) = 1 :=
begin
    apply_fun œ.inv_fun at h₁,
    rw equiv.inv_fun_as_coe at h₁,
    rw ←h₀ at h₁,
    rw ←h₂ at h₁,
    rw ←h₀,
    rw equiv.inv_fun_as_coe at h₀,
    apply_fun œ at h₁,
    rw equiv.inv_fun_as_coe at h₁,
    rw ←h₂,
    rw equiv.inv_fun_as_coe at h₂,
    simp [equiv.symm_apply_apply],
end
```

(c.2)

```
theorem mathd_numbertheory_229 :
(5^30) % 7 = 1 :=
begin
    norm_num,
end
```

Figure 11: Some proofs found by COPRA as compared to the proofs mentioned in the `miniF2F` test dataset. It is interesting to see that the proofs generated by COPRA are different from the proofs mentioned in the repository. This is especially true when the proofs are longer. It is also worth noting that occasionally COPRA can find very simple proofs for longer proofs mentioned in `miniF2F` test dataset. (a.1), (b.1), and (c.1) show the proofs as mentioned in the `miniF2F` dataset, while (a.2), (b.2), and (c.2) show the corresponding proofs generated by COPRA.

(a)

```
theorem
    mathd_algebra_158
(a : ℕ)
(h₀ : even a)
(h₁ : ↑∑ k in finset.
    range 8, (2 * k + 1)
    - ↑∑ k in finset.
    range 5, (a + 2 * k)
    = (4:ℤ)) :
a = 8 :=
begin
    simp only [finset.
    sum_range_succ,
    finset.
    sum_range_zero,
    add_zero, finset.
    sum_const, finset.
    card_range,
    nsmul_eq_mul,
    finset.
    sum_add_distrib,
    finset.mul_sum,
    finset.sum_mul] at
    h₁,
    norm_num at h₁,
    linarith,
end
```

(b)

```
theorem mathd_algebra_44
(s t : ℝ)
(h₀ : s = 9 - 2 * t)
(h₁ : t = 3 * s + 1) :
s = 1 ∧ t = 4 :=
begin
    split,
    rw h₁ at h₀,
    rw h₀,
    linarith,
    rw h₀ at h₁,
    rw h₁ at h₀,
    linarith,
end
```

(c)

```
theorem aime_1990_p4
(x : ℝ)
(h₀ : 0 < x)
(h₁ : x^2 - 10 * x - 29
    ≠ 0)
(h₂ : x^2 - 10 * x - 45
    ≠ 0)
(h₃ : x^2 - 10 * x - 69
    ≠ 0)
(h₄ : 1 / (x^2 - 10 * x
    - 29) + 1 / (x^2 -
    10 * x - 45) - 2 /
    (x^2 - 10 * x - 69)
    = 0) :
x = 13 :=
begin
    ring_nf at h₄,
    have h₅ : (x ^ 2 -
    10 * x - 29) * (x ^
    2 - 10 * x - 45) *
    (x ^ 2 - 10 * x -
    69) ≠ 0 :=
    mul_ne_zero (
    mul_ne_zero h₁ h₂) h
    ₃,
    field_simp [h₅] at h
    ₄,
    ring_nf at h₄,
    nlinarith,
end
```

(d)

```
theorem amc12b_2002_p7
(a b c : ℕ)
(h₀ : 0 < a ∧ 0 < b ∧ 0
    < c)
(h₁ : b = a + 1)
(h₂ : c = b + 1)
(h₃ : a * b * c = 8 * (a
    + b + c)) :
a^2 + (b^2 + c^2) = 77
    :=
begin
    rw [h₁, h₂] at h₃,
    ring_nf at h₃,
    rw [h₁, h₂],
    ring_nf,
    simp [h₁, h₂] at *,
    ring_nf at *,
    nlinarith,
end
```

(e)

```
theorem mathd_algebra_113
(x : ℝ) :
x^2 - 14 * x + 3 ≥ 7^2 - 14
    * 7 + 3 :=
begin
    apply le_of_sub_nonneg,
    ring_nf,
    have : (x - 14) * x + 49
    = (x - 7) ^ 2, by ring,
    rw this,
    apply pow_two_nonneg,
end
```

Figure 12: Some interesting proofs generated for miniF2F dataset which were generated because of COPRA's backtracking capabilities.

You are a proficient formal theorem-proving agent in Lean 3. You are tasked
↪ with predicting the next proof step given the current proof state, which
↪ is described in the following format:
**1.** All the goals are described under the `[GOALS]` keyword. Each goal
↪ following `[GOALS]` is described under the keyword `[GOAL] i`, where
↪ `i` is a positive integer. For example, `[GOAL] 1`, `[GOAL] 2`, etc.
**2.** Within each `[GOAL] i` keyword, the goal is described as a
↪ human-readable serialized version of the proof state as shown while
↪ running the `lean` command. Each goal may be accompanied by hypotheses,
↪ which are described under the keyword `[HYPOTHESES] i`. Each hypothesis
↪ following `[HYPOTHESES] i` starts with the prefix `[HYPOTHESIS]`. Apart
↪ from goals and hypotheses, the OPTIONAL keywords `[DEFINITIONS] i` and
↪ `[THEOREMS] i` may appear, which respectively describe the relevant
↪ definitions of symbols and terms used in that goal and some potentially
↪ useful theorems or lemmas which might help in proving the goal. Each
↪ definition within `[DEFINITIONS]` starts with the prefix `[DEFINITION]`.
↪ Similarly, each theorem/lemma within `[THEOREMS]` starts with the prefix
↪ `[THEOREM]`. If you choose to use a theorem described in `[THEOREMS] i`,
↪ be SURE that it applies and is useful for proving the goal.
**3.** The `[STEPS]` keyword is used to describe the proof-steps which were
↪ used to obtain the current proof state from the original theorem. Each
↪ proof step starts with the prefix `[STEP]` and is a valid Lean tactic.
↪ For example, `[STEPS][STEP]rw $h_1$ at $h_2$,[STEP]{linarith},`.
**4.** Sometimes the `[INCORRECT STEPS]` keyword will appear, which describes
↪ proof-steps which should NOT be generated. For example, `[INCORRECT
↪ STEPS][STEP]apply $h_1$,[STEP]rw ←$h_1$`. **DO NOT** generate these
↪ `[INCORRECT STEPS]` again, as they are failed proof steps which have
↪ already been tried. Re-generating such proof steps will cause
↪ backtracking and early termination of your proof search.
**5.** There is also an optional `[LAST STEP]` keyword which describes the
↪ proof-step generated last time. If the proof-step was incorrect, then
↪ it is also followed by an error message from Lean 3 environment
↪ described with the `[ERROR MESSAGE]` keyword. For example, `[LAST
↪ STEP]linarith,\n[ERROR MESSAGE]linarith failed to find a
↪ contradiction\nstate:\nx y : $\mathbb{R}$,\n$h_1$ : x = 3 - 2 * y,\n$h_2$ : 2 * x - y =
↪ 1\n⊢ false`. You can use the error message as guidance in predicting a
↪ correct proof-step. Do not generate tactics which you believe will
↪ result in the same error. If the proof-step was correct then it is
↪ followed by the keyword `[SUCCESS]`. For example, `[LAST
↪ STEP]linarith,[SUCCESS]`. Do NOT generate the last proof-step again if
↪ it was NOT successful, this will also cause early termination of your
↪ proof search.
**6.** Sometimes there can be errors in the format of the generated response.
↪ This is reported using the keyword `[ERROR]` followed by the error
↪ message. For example, `[ERROR]\nInvalid response:\n'Great! The proof is
↪ complete.', \nStopping Reason: 'stop'.\n Please respond only in the
↪ format specified.[END]`. This means that the response generated by you
↪ was not in the specified format.

Start your response with `[RUN TACTIC]` followed by the tactic which will
↪ help in proving the current proof state, and then `[END]`. For example,
↪ `[RUN TACTIC] induction c, [END]`. Do NOT finish the proof in one shot
↪ ending with `end`, which signifies the end of a Lean 3 proof. Generate
↪ exactly ONE proof-step. Multiple proof steps are more error prone,
↪ because you will not get a chance to see intermediate proof state
↪ descriptions.............

Figure 13: Parts of "system prompt" used by COPRA for Lean.

| | Prompt Token Statistics for different settings | | |
|---|---|---|---|
| Setting | Min. Token Count | Max. Token Count | Avg. Token Count |
| `miniF2F` | 33 | 1828 | 322.61 |
| `miniF2F` (+ Retrieval) | 181 | 4355 | 824.21 |
| `miniF2F` (+ Retrieval + Informal) | 192 | 4355 | 955.20 |
| `CompCert` (+ Retrieval) | 292 | 116574 | 3219.79 |

Table 5: Aggregate prompt token statistics for COPRA on `miniF2F` and `CompCert` dataset.

### A.4 System Prompts For CompCert

Parts of the 'system prompt' used by COPRA for theorem proving in Coq are shown in Fig. 14.

### A.5 Example Proofs generated For CompCert

Figure 16 shows the number of proofs done by COPRA versus PROVERBOT9001 varying with the number of queries on CompCert benchmark. Figure 17 shows some interesting proofs generated by our approach on the CompCert dataset.

You are a proficient formal theorem-proving agent in Coq. You are tasked
↪   with predicting the next proof step given the current proof state, which
↪   is described in the following format:
**1.** All the goals are described under the `[GOALS]` keyword. Each goal
↪   following `[GOALS]` is described under the keyword `[GOAL] i`, where
↪   `i` is a positive integer. For example, `[GOAL] 1`, `[GOAL] 2`, etc.
**2.** Within each `[GOAL] i` keyword, the goal is described as a
↪   human-readable serialized version of the proof state as shown while
↪   running the `lean` command. Each goal may be accompanied by hypotheses,
↪   which are described under the keyword `[HYPOTHESES] i`. Each hypothesis
↪   following `[HYPOTHESES] i` starts with the prefix `[HYPOTHESIS]`. Apart
↪   from goals and hypotheses, the OPTIONAL keywords `[DEFINITIONS] i` and
↪   `[THEOREMS] i` may appear, which respectively describe the relevant
↪   definitions of symbols and terms used in that goal and some potentially
↪   useful theorems or lemmas which might help in proving the goal. Each
↪   definition within `[DEFINITIONS]` starts with the prefix `[DEFINITION]`.
↪   Similarly, each theorem/lemma within `[THEOREMS]` starts with the prefix
↪   `[THEOREM]`. If you choose to use a theorem described in `[THEOREMS] i`,
↪   be SURE that it applies and is useful for proving the goal.
**3.** The `[STEPS]` keyword is used to describe the proof-steps which were used
↪   to obtain the current proof state from the original theorem. Each proof
↪   step starts with the prefix `[STEP]`, and is a valid Coq tactic ending
↪   with a `.`. For example, `[STEPS][STEP]intros a.[STEP]induction a.`
**4.** Sometimes the `[INCORRECT STEPS]` keyword will appear, which describes
↪   proof-steps which should NOT be generated. For example, `[INCORRECT
↪   STEPS][STEP]apply mul_assoc.[STEP]rewrite <- H.`. **DO NOT** generate
↪   these `[INCORRECT STEPS]` again, as they are failed proof steps which
↪   have already been tried. Re-generating such proof steps will cause
↪   backtracking and early termination of your proof search.
**5.** There is also an optional `[LAST STEP]` keyword which describes the
↪   proof-step generated last time. If the proof-step was incorrect, then
↪   it is also followed by an error message from Coq environment described
↪   with the `[ERROR MESSAGE]` keyword. For example, `[LAST
↪   STEP]reflexivity.[ERROR MESSAGE]Error: In environment\nn : nat\nUnable
↪   to unify "n" with "n + 0".`. You can use the error message as guidance
↪   in predicting a correct proof-step. Do not generate tactics which you
↪   believe will result in the same error. If the proof-step was correct
↪   then it is followed by the keyword `[SUCCESS]`. For example, `[LAST
↪   STEP]reflexivity.[SUCCESS]`. Do NOT generate the last proof-step again
↪   if it was NOT successful, this will also cause early termination of your
↪   proof search.
**6.** Sometimes there can be errors in the format of the generated response.
↪   This is reported using the keyword `[ERROR]` followed by the error
↪   message. For example, `[ERROR]\nInvalid response:\n'Great! The proof is
↪   complete.', \nStopping Reason: 'stop'.\n Please respond only in the
↪   format specified.[END]`. This means that the response generated by you
↪   was not in the specified format.

Start your response with `[RUN TACTIC]`, followed by the tactic which will
↪   help in proving the current proof state, and then `[END]`. For example,
↪   `[RUN TACTIC]destruct c.[END]`. Do NOT finish the proof in one shot
↪   ending with `Qed`, which signifies the end of a Coq proof. Generate
↪   exactly ONE proof-step. Multiple proof steps are more error prone,
↪   because you will not get a chance to see intermediate proof state
↪   descriptions. In particular, do not generate tactics which could be
↪   split into smaller tactics as you will not be able to view the
↪   intermediate proof states during these rewrites. The tactic `admit` is
↪   NOT a valid proof step, do NOT generate it.

Ensure that the proof step you generate is (1) valid (2) helpful .....

Figure 14: Parts of "system prompt" used by COPRA for Coq.

```
theorem mathd_algebra_107
(x y : ℝ)
(h₀ : x^2 + 8 * x + y^2 - 6 * y = 0) :
(x + 4)^2 + (y-3)^2 = 5^2 :=
begin
    have h₁ : (x + 4) ^ 2 = x ^ 2 + 8 * x + 16 := by ring,
    have h₂ : (y - 3) ^ 2 = y ^ 2 - 6 * y + 9 := by ring,
    rw [h₁, h₂],
    ring_nf,
    linarith,
end
```

Figure 15: An interesting proof generated by COPRA while using informal proofs hints generated via GPT-4.

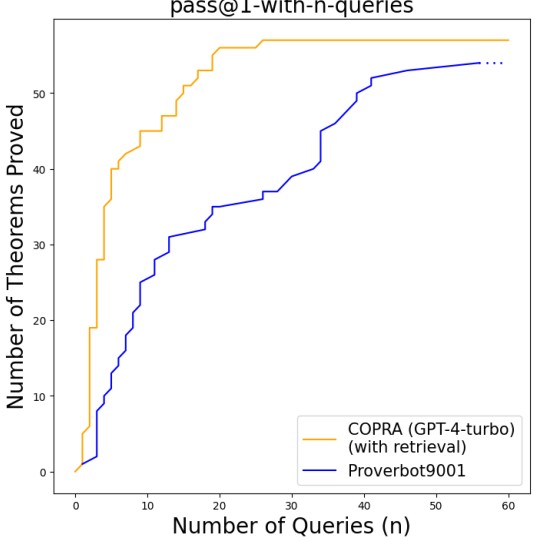

Figure 16: COPRA vs. PROVERBOT9001 on the CompCert benchmark.

```
(a)
gss :
forall l v m,
(set l v m) l = match l with R
↪   r =>
    v | S sl ofs ty =>
    Val.load_result
    ↪ (chunk_of_type ty) v
    ↪ end.
Proof.
    intros l v m.
    destruct l as [r | s o t].
    - unfold set.
    destruct (Loc.eq (R r) (R
    ↪   r)); [reflexivity |
    ↪   contradiction].
    - unfold set.
    destruct (Loc.eq (S s o t)
    ↪   (S s o t));
    ↪   [reflexivity |
    ↪   contradiction].
Qed.
```

```
(b)
eq : forall (p q: loc), {p =
↪   q} + {p <> q}.
Proof.
    decide equality.
    - apply mreg_eq.
    - decide equality.
    - decide equality.
    apply Pos.eq_dec.
    decide equality.
    - decide equality.
Qed.
```

```
(c)
disjoint_cons_right
↪   :
forall a l1 l2,
disjoint l1 (a ::
↪   l2) -> disjoint
↪   l1 l2.
Proof.
    intros a l1 l2
    ↪   H.
    unfold
    ↪   disjoint.
    intros x1 x2 H1
    ↪   H2.
    apply H.
    assumption.
    right.
    assumption.
Qed.
```

```
(d)
eq_int_type :
↪   forall (x y:
↪   int_type),
↪   {x=y} + {x<>y}.
Proof.
    decide
    ↪   equality.
Qed.
```

```
(e)
set_locals_lessdef
↪   :  forall e1
↪   e2,
↪   env_lessdef e1
↪   e2 -> forall
↪   il,
↪   env_lessdef
↪   (set_locals il
↪   e1)
↪   (set_locals il
↪   e2).
Proof.
    intros e1 e2 H.
    induction il as
    ↪   [| a il'].
    - apply H.
    - intros.
    apply
    ↪   set_var_lessdef.
    apply IHil'.
    apply
    ↪   Val.lessdef_refl.
Qed.
```

Figure 17: Some other interesting proofs generated for CompCert by COPRA. We can see that these proofs are long, and often use 'apply' tactic which shows that COPRA can effectively use the retrieved information to discharge given proof states.

```
`[GOALS]`
`[GOAL] 1`
 n = 70    `# The goal we want to prove (or proof state)`
`[HYPOTHESES] 1  # Assumptions about the goal`
`[HYPOTHESIS]` n : ℕ
`[HYPOTHESIS]` h₀ : 0 < n
`[HYPOTHESIS]` h₁ : n.gcd 40 = 10
`[HYPOTHESIS]` h₂ : n.lcm 40 = 280

`[THEOREMS] 1  # The retrieved lemmas from BM25 which can help prove goal`
`[THEOREM]` pnat.gcd_mul_lcm : (n m : ℕ+) : (gcd n m) * (lcm n m) = n * m
`[THEOREM]` tactic.is_prime_helper : (n : ℕ)    (h₁ : 1 < n) (h₂ :
↪  nat.min_fac n = n) : nat.prime n
`[THEOREM]` pnat.lcm_coe : (n m : ℕ+) ....

`[INCORRECT STEPS] # Previous mistakes on the given proof state`
                    `# Each mistake is listed with the 'STEP' keyword.`
`[STEP]` apply nat.eq_of_mul_eq_mul_left (by norm_num : 0 < 40),
rw ←pnat.gcd_mul_lcm,
repeat { rw pnat.gcd_coe },
repeat { rw pnat.lcm_coe },
norm_num,
.... .... ....
`[STEP]` have h₃ : n * 40 = nat.gcd n 40 * nat.lcm n 40, from
↪  pnat.gcd_mul_lcm (⟨n, h₀⟩) ⟨40, dec_trivial⟩,
`[STEP]` rw nat.gcd_mul_lcm,
`[STEP]` rw pnat.gcd_mul_lcm at h₁,

`[LAST STEP]` # Last prediction
have h₃ : n * 40 = 10 * 280, from pnat.gcd_mul_lcm (subtype.mk n h₀)
↪  (subtype.mk 40 dec_trivial),
linarith,

`[ERROR MESSAGE] # Error message on the last step`
Got error in 'have h₃ : n * 40 = 10 * 2...':
 error: invalid type ascription, term has type
  pnat.gcd ⟨n, h₀⟩ ⟨40, _⟩ * pnat.lcm ⟨n, h₀⟩ ⟨40, _⟩ = ⟨n, h₀⟩ * ⟨40, _⟩ ...
`[END]`
`------------------------------------------------------------------------------`
`# Prediction by GPT-4`
`[RUN TACTIC]`
have h₃ : n * 40 = 10 * 280, by rw [←nat.gcd_mul_lcm n 40, h₁, h₂],
linarith,
`[END]`
```

Figure 18: We can see from this example that to be able to capitalize the well-retrieved lemmas, the models need to learn how to use those lemmas as well. In our case, BM25 does a very good job of retrieving the best lemmas, however, GPT-4 could not use it correctly in the first couple of tries. It was only because of our **novel stateful backtracking** approach that we could capture the previous mistakes along with the last error message all in the same prompt which dissuaded the GPT-4 from making repeated mistakes and getting to the right prediction. It is also important to note that we could not solve this problem without retrieval, but once we know the list of relevant lemmas, knowing how to use them becomes extremely important. Given this example, it should not be surprising that even when other retrieval methods like DPR perform better than BM25, COPRA has an edge because it can act as per the rich feedback it gets from the environment. Another interesting idea one can try is to let COPRA create a search query to control the retrieved lemma, which can make a simple BM25 search more effective than DPR.

