# OpenReview forum: "An In-Context Learning Agent for Formal Theorem-Proving"
_colmweb.org/COLM/2024/Conference — COLM_

### Official Review · Reviewer_BSYj · 2024-05-09

**Rating:** 5
**Confidence:** 4
**Ethics Flag:** 2

**Summary:**

This paper presents an in-context learning agent for formal theorem proving, where prompts will be revised according to the error messages from the formal environment. The effectiveness of the proposed approach named COPRA has been validated on both Lean (miniF2F) and Coq (CompCert).

**Questions To Authors:**

page 3, 'we have an efficient symbolic procedure that can check this relationship for any pair of states': could elaborate a bit on how to check if one proof obligation is stronger than another?

**Reasons To Accept:**

Strength: the paper is well written and easy to follow. I especially appreciate that the authors attempt to validate their idea on two diverse datasets (miniF2F and CompCert) from two systems (Lean and Coq). I also like the introduction of the metric that incorporates the number of queries to the LLMs.

**Reasons To Reject:**

Weakness: the effectiveness of the proposed approach is still a bit questionable, considering the ReProver baseline in Lean uses a much much smaller model and Proverbot9001 in Coq is not even based on language models and uses way less resources. It is also concerning to see the performance of Copra drops significantly when equipped with a less capable model (e.g., CodeLlamma and GPT-3.5). Admittedly, domain-specific fine-turning/training may explain some of the discrepancy between Copra (CodeLlamma/GPT-3.5) and ReProver, but I believe we still need to see the effectiveness of Copra when equipped with less capable models.

---

> ### Author Rebuttal · Authors · 2024-05-28
>
> Thank you for your review. We address your main points below.
>
>
> **Question**: page 3, ‘we have an efficient ….’: Elaborate a bit on how to check if one proof obligation is stronger than another.
>
> **Answer**: The line you quoted does not appear in our submission. We presume you are referring to our discussion of Progress Checks on Page 4. In that section, we describe a procedure that is used to eliminate some straightforward cases where we can compare the difficulty of two proof states. Summarized, these cases are when goals match exactly and one set of hypotheses contains the other. We are happy to elaborate further on this if you have further questions.
>
>
> **Question**: Comparison with ReProver and Proverbot9001.
>
> **Answer**: Our approach is faster than ReProver both in terms of wall-clock time as well as the number of queries. We used Proverbot9001 because it is the best model for the CompCert dataset. Given the use of extra signals like error feedback, and proof history, we induce faster proof completion while searching for proofs. It is important to note that our approach is completely in-context and doesn’t need any domain-specific training data. Being We have shown that it works on both miniF2F and CompCert, while ReProver and Proverbot are limited to specific languages and the specific domains from which their training data is drawn.
>
>
> **Concern**:  We need to see the effectiveness of Copra when equipped with less capable models.
>
> **Answer**:   Prior work has shown that some of the in-context reasoning capabilities of LLM agents only kick in when the model is of GPT-4 scale. For example, Voyager, which used LLM agents to play Minecraft, needed GPT-4 as the actor. We do not believe that our use of GPT-4 should count against Copra, especially as in the future, GPT-4-scale models will get cheaper.
>
> Also note that smaller models have smaller context windows and sometimes cannot fit in all feedback signals. Generating the feedback data (error signals, proof history, etc.) to train the smaller models is harder because we don’t have data with error signal annotation to begin with. Our use of ICL/GPT-4 lets us get around the need to train on feedback data.

---

> > ### Comment · Reviewer_BSYj · 2024-06-05
> >
> > I very much appreciate the authors' clarification. However, I am still concerned about the comparison between COPRA and the baselines (ReProver and Proverbot9001), where GPT-4 has been used to compete against small (or even none-LM based) models. Both ReProver and Proverbot9001 can be deployed locally, while GPT-4 is still too expensive. I am worried the advantages of COPRA might be diminished once ReProver and Proverbot9001 use stronger models. By the way, you can consider Graph2Tac (https://arxiv.org/abs/2401.02949) as a baseline in Coq, which is more recent and easier to adapt to LLM settings. Additionally, is the significant performance drop from GPT-4 to GPT-3.5 solely a result of the difference in context width? While I am open to the use of GPT-4, I would like to see a more comprehensive discussion on the specific model requirements for COPRA, as well as the feasibility of implementing COPRA using open-source or more academically accessible models.  Given the reasons outlined above, I will keep my rating unchanged.

---

> > > ### Author Response · Authors · 2024-06-05
> > >
> > > Thank you for your comments. We would like to address some of those below:
> > >
> > > **Question:** Comparison between COPRA and baselines with smaller models.
> > >
> > > **Answer:** ReProver and Proverbot9001 are finetuning approaches and require training before use. For COPRA style use, it will need training on the feedback (error signals) data which is hard to generate. In general, in NLP, there has been a shift from finetuning to ICL. The power of ICL in theorem-proving is, in our opinion, a fundamental topic of interest to the COLM community. Open models are improving rapidly -- GPT-4 scale open models are not necessarily that far away.
> > >
> > > **Question:** Is the significant performance drop from GPT-4 to GPT-3.5 solely a result of the difference in context width?
> > >
> > > **Answer:** We believe that the drop from GPT-4 to GPT-3.5 is possibly because of the context window but also possibly because of the differences in the larger pretraining set. That said, we have thoroughly analyzed our results and found no evidence of leakage. Therefore, there seems to be a jump in the theorem-proving capabilities, and this is an intriguing phenomenon that the COLM community is well-positioned to study more deeply.
> > >
> > > **Question:** Feasibility of implementing COPRA using open-source or more academically accessible models?
> > >
> > > **Answer:** We evaluated COPRA on CodeLlama 2 (see Table 1) which is open for academic usage. This proves that our approach is agnostic of the underlying model. We show that using COPRA we could boost the performance of few-shot CodeLlama from 0% to 5.73%, which means the novel stateful backtracking search helps regardless of the underlying model. However, the performance is not comparable to GPT-4 because CodeLlama is a less powerful model. We would like to reiterate that open models at the level of GPT-4 are likely to be available in the future due to the rapid progress of the open-source community. For example, we can potentially incorporate a baseline where Llama 3 is used as the foundation model driving COPRA in subsequent revisions.
> > >
> > > We would like to thank you for pointing out Graph2Tac, we were not aware of this work and we will cite it. We note that this is a fairly recent work, and we can include it as a baseline in subsequent revisions.

---

### Official Review · Reviewer_eJQk · 2024-05-11

**Rating:** 6
**Confidence:** 4
**Ethics Flag:** 1

**Summary:**

This paper introduces a new search algorithm, COPRA, to tackle the problem of theorem proving. COPRA uses a neat search algorithm, (branded as an "agentic approach") that incorporates elements such as execution feedback and a state comparison heuristic. Because it only uses in-context interactions, it is generally applicable to various domains and shows promising results on miniF2F and CompCert. Overall, the paper is clearly written, interesting, and the method is effective.

As I detail below, my two concerns are as follows. First, there is a limited evaluation, as the authors only test on two datasets. In general, this is fine, but I suspect miniF2F may be contaminated, as the problems are likely also on many mathematical websites. An additional dataset such as LeanDojo benchmark would be more convincing. Second, the method still falls behind SoTA, and it would be more convincing if the authors incorporated fine-tuned language models into their framework to achieve a better performance.

**Questions To Authors:**

- What is the effect of including proof feedback? Can models like GPT-4 really learn from feedback? I would be interested in seeing an ablation study where error feedback is omitted when prompting (but still providing it the feedback that things are incorrect).
- Have the authors tried combining small, powerful models such as Llemma or ReProver with the feedback abilities of GPT-4 to further reduce the wall-clock time of proving theorems? As the authors mention, I don't find it unlikely that small models are relatively sufficient for theorem-proving, and I wonder if replacing some of the GPT-4 interactions with Llemma or ReProver would lead to better results.
- The boost from using informal statements is less than I would have expected, especially in light of the success of DSP-like methods. Could the authors comment on this?
- Can the LLM also provide feedback on the retrieval, to identify whether a retrieval was helpful for the next step?
- Suggestion: I would consider bringing an example of how COPRA works (something like Figure 8) into the main text, which would help me understand the algorithm better than the formalism on pages 3-4.

**Reasons To Accept:**

- COPRA is a new, agent-based search approach for tackling the problem of formal theorem proving. Their method beats strong baselines such as ReProver and Llemma, even without retrieval.
- The search procedure is intuitive and straightforward. It shows good results compared to GPT-4, ReProver, and Llemma.
- To my knowledge, this is the first work to incorporate theorem prover feedback into the process without fine-tuning.
- The authors provide a relatively comprehensive set of experiments, comparing not just query count but also wall-clock time (which is important because ReProver is likely much faster than GPT-4 for every single query). They also ablate backtracking, informal proofs, and retrieval and also give an analysis of contamination.

**Reasons To Reject:**

- While the authors give a contamination analysis on miniF2F, I don't find it unlikely that elements such as informal proofs or even the formal proofs themselves may be leaked somewhere on GitHub, not necessarily just in the miniF2F-test repository. This could give it an advantage over its baselines.
- The work only evaluates two datasets. In light of this and the previous point, an additional dataset would be more convincing: for example, if the method can beat ReProver with retrieval at a subset of the LeanDojo benchmark.
- As the baselines included in the paper are mostly simple fine-tuned approaches, the results still fall behind SoTA. For example, HTPS achieve 35% with supervised fine-tuning, and 41% using their search algorithm, while COPRA is still at 31%. I acknowledge the in-context nature of COPRA compared to the fine-tuned setting of many of these works, but I believe the authors should have some comparison to methods like these, which have a better performance.

---

> ### Author Rebuttal · Authors · 2024-05-28
>
> Thank you for your thoughtful review! Below are the responses to your queries.
>
>
> **Q**: Impact of omitting error feedback.
>
>
> **A**: These would indeed be interesting ablations, and we would be happy to add these ablations to the paper.
>
> **Q**: Limited evaluation and possible contamination.
>
> **A**: While including additional eval-sets would certainly strengthen our paper, we already go beyond most prior work on AI for theorem-proving by handling multiple languages (Coq, Lean) and two distinct application domains (competition mathematics and software verification). We could add an evaluation on a subset of the LeanDojo benchmark if necessary.
>
> For contamination issues, we performed a deep assessment of the proofs found by COPRA, generally showing no duplication (Table 4 in App.). No notable results popped up when we looked up the (longer) proofs generated by COPRA.
>
> **Q**: Combining smaller models with feedback abilities of GPT-4.
>
> **A**: We tried out two smaller models -- CodeLlama and GPT-3.5 -- for in-context tactic selection. We found to often fail to adhere to the prompt format. However, using a larger-scale model as a critic for such a smaller model -- and combining finetuned and in-context models --are  interesting ideas for future work.
>
>
> **Q**: Limited boost from informal proofs and results still fall behind SoTA
>
> **A**: HTPS and DSP use pass@64 and pass@100 respectively, while we use pass@1. SoTA depends on higher value of k in pass@k.
> In the DSP paper (https://arxiv.org/pdf/2210.12283), from Figure 3 we can see that their pass@1 for combined miniF2F test + valid set is around 70. Thus, the test set number should be around 35 (test and valid set are same size). While we could do 75 proofs (30.74%) on the test set alone.
>
>
> **Q**: Can LLM also provide feedback on the retrieval?
>
> **A**:  In our experiments, the LLM does not. However, using the LLM in this way is an interesting idea for future work.
>
> Thank you for the suggestion regarding Fig. 8. We will move it to the main paper.

---

> > ### Author Response · Authors · 2024-06-06
> >
> > Dear Reviewer,
> >
> > Please let us know if you have any questions or clarifications about our response, we would be happy to address them.

---

### Official Review · Reviewer_Ub88 · 2024-05-21

**Rating:** 6
**Confidence:** 4
**Ethics Flag:** 1

**Summary:**

This paper presents an in-context learning agent called COPRA for formal theorem-proving with Lean and Coq. This is achieved by calling general-purpose models like GPT-4 with the help of frameworks like LeanDoJo. During the tactics generation/proof search process, feedback from execution and search history is integrated to get better performance. On selected benchmarks, COPRA outperforms few-shot invocations of GPT-4 (without the help of theorem-proving environment) and fine-tuned baselines like ReProver.

**Reasons To Accept:**

Theorem proving is a unique problem because you can get feedback on whether you've solved it by checking for remaining goals. Additionally, you receive feedback on intermediate execution through frameworks like LeanDoJo, which prevents incorrect tactics. This is an interesting problem setting to integrate the feedback in an self-improving way.

The analysis is thorough. Experiments compared the accuracy with few-shot LLM queries and fine-tuned baselines. The qualitative analysis is also interesting.

I appreciate the discussion on test-set memorization concerns, as addressing memorization issues in black-box large language models is crucial.

**Reasons To Reject:**

The novelty of this paper is limited. Existing models like ReProver already suggest tactics given goals/premises, and it's known that powerful LLMs like GPT-4 have strong general capabilities, so simply integrating them isn't a major contribution.

The paper could benefit from more in-depth analysis. For example, what problems can GPT-4 solve that ReProver cannot? Since ReProver is trained on mathlib3, what logic types/tactics does it master better? Conversely, what logic types/tactics does GPT-4 master better, and why (could it be due to GPT-4 being trained on a broader range of web-scale data beyond mathlib3)?

Stylistic suggestions:
Figure 2 would look better if (a) and (c) were aligned vertically.
Figure 3 could be improved by placing the notations at the top of the algorithm description, which would enhance reader understanding and align with (more) standard practices (see Algorithm 1 of https://arxiv.org/pdf/2310.10625).

---

> ### Author Rebuttal · Authors · 2024-05-28
>
> Thank you for your valuable feedback. We address your main points below.
>
>
> **Concern**: The novelty is limited, given ReProver already suggests tactics given goals/premises.
>
> **Answer**: Our proof search procedure goes beyond Reprover in using additional signals not present at Reprover's training time, including error messages, memoization of proof search history, and additionally retrieved lemmas. Also, Copra handles both Coq and Lean, and we demonstrate its applicability in two quite different domains: competition mathematics and software verification. By contrast, other AI-based approaches to formal mathematics tend to be restricted to specific proof environments and applications. Also, compared to ReProver and Proverbot, we find that COPRA is more query-efficient. Our wall-clock time analysis also shows that the COPRA can find proofs faster than ReProver. Finally, we believe the use of in-context learning in difficult reasoning tasks is a subject of intrinsic interest to the language-models community. Copra is the first system to exploit in-context learning in formal theorem-proving.
>
>
>
> **Question**: Examples of problems that ReProver cannot solve but COPRA could. In-depth analysis of tactics used in proofs.
>
>
> **Answer**: Figure 6 shows one example where ReProver fails while COPRA succeeds. Figure 9 shows the breakdown of theorems proved in various categories by REPROVER and COPRA, we can see that COPRA proves more than ReProver in almost all categories (while in some categories they both fail like induction proofs). Particularly, COPRA is generally more capable of handling problems in the algebra category. Table 4 (in the appendix) sheds some light on the types of tactics used by COPRA, however, we can do a more in-depth analysis beyond the basic tactics in subsequent revisions.
>
> Finally, thank you for the stylistic suggestions. We will incorporate them into the next version of the paper.

---

> > ### Comment · Reviewer_Ub88 · 2024-06-04
> > **Thanks for the rebuttal**
> >
> > I acknowledge that your work includes additional elements (such as error messages) compared to ReProver. However, these are standard practices for self-improvement, which is why I commented on the lack of novelty.
> >
> > Regarding Figure 6, I'm finding it difficult to comprehend. ReProver also employs retrieval (Dense Passage Retriever) during the proof search process. Could you explain why a potentially weaker retriever like BM25 achieves better performance than ReProver?

---

> > > ### Author Response · Authors · 2024-06-05
> > >
> > > Thank you for your comment. We would like to address your concerns as follows:
> > >
> > > **Comment:** “Standard practice for self-improvement that leads to lack of novelty.”
> > >
> > > **Response:** We go beyond just including the error response. The error message is incorporated with a stateful backtracking search. Statefulness helps avoid repeating similar mistakes and helps prune the search space further. Apart from the error message on the last step, the proof steps that were incorrect for the particular state are shown in the prompt, this reduces the search space dramatically, and triggers backtracking if the LLM chooses to repeat the same incorrect step. The stateful tracking of errors allows us to identify proof steps that eventually fail even while executing immediately which further helps in backtracking early and changing the direction of proof. We have included an example of this in the response to your next comment.
> > >
> > > **Comment:** “ReProver uses DPR why BM25 works better in COPRA than ReProver”
> > >
> > > **Response:** Retrieval helps get the useful lemmas needed to complete a proof, but the tactic prediction model needs to know the right way of using them. The model may fail to predict the right arguments for the lemma, and the proof search may fail despite the retrieval being correct. The ReProver paper doesn’t perform an ablation on how the quality of the retriever impacts its ability to do proofs. It studies the quality of the DPR vs BM25 retriever in terms of MRR but doesn’t have any ablations where ReProver uses BM25 instead of DPR for proof search, so it is hard to know the impact of the quality of retrieved results on the overall proof search.
> > >
> > >
> > > For the example in Fig. 6, we had the following prompt just before GPT correctly predicted the next proof step:
> > >
> > > ```
> > > [GOALS]
> > > [GOAL] 1
> > > n = 70   # The goal we want to prove (or proof state)
> > > [HYPOTHESES] 1  # Assumptions about the goal
> > > [HYPOTHESIS] n : ℕ
> > > [HYPOTHESIS] h₀ : 0 < n
> > > [HYPOTHESIS] h₁ : n.gcd 40 = 10
> > > [HYPOTHESIS] h₂ : n.lcm 40 = 280
> > >
> > >
> > > [THEOREMS] 1  # The retrieved lemmas from BM25 which can help prove goal
> > > [THEOREM] pnat.gcd_mul_lcm : (n m : ℕ+) : (gcd n m) * (lcm n m) = n * m
> > > [THEOREM] tactic.is_prime_helper : (n : ℕ)   (h₁ : 1 < n) (h₂ : nat.min_fac n = n) : nat.prime n
> > > [THEOREM] pnat.lcm_coe : (n m : ℕ+) ....
> > >
> > >
> > > [INCORRECT STEPS] # Previous mistakes on the given proof state
> > >                   # Each mistake is listed with the ‘STEP’ keyword.
> > > [STEP] apply nat.eq_of_mul_eq_mul_left (by norm_num : 0 < 40),
> > > rw ←pnat.gcd_mul_lcm,
> > > repeat { rw pnat.gcd_coe },
> > > repeat { rw pnat.lcm_coe },
> > > norm_num,
> > > .... .... ....
> > > [STEP] have h₃ : n * 40 = nat.gcd n 40 * nat.lcm n 40, from pnat.gcd_mul_lcm (⟨n, h₀⟩) ⟨40, dec_trivial⟩,
> > > [STEP] rw nat.gcd_mul_lcm,
> > > [STEP] rw pnat.gcd_mul_lcm at h₁,
> > >
> > >
> > > [LAST STEP] # Last prediction
> > > have h₃ : n * 40 = 10 * 280, from pnat.gcd_mul_lcm (subtype.mk n h₀) (subtype.mk 40 dec_trivial),
> > > linarith,
> > >
> > >
> > > [ERROR MESSAGE] # Error message on the last step
> > > Got error in 'have h₃ : n * 40 = 10 * 2...':
> > >  error: invalid type ascription, term has type
> > >   pnat.gcd ⟨n, h₀⟩ ⟨40, _⟩ * pnat.lcm ⟨n, h₀⟩ ⟨40, _⟩ = ⟨n, h₀⟩ * ⟨40, _⟩ ...
> > > [END]
> > > ```
> > >
> > > ```
> > > # Prediction by GPT-4
> > > [RUN TACTIC]
> > > have h₃ : n * 40 = 10 * 280, by rw [←nat.gcd_mul_lcm n 40, h₁, h₂],
> > > linarith,
> > > [END]
> > > ```
> > >
> > > It is easy to see that to be able to capitalize the well-retrieved lemmas, the models need to learn how to use those lemmas as well. In our case, BM25 does a very good job of retrieving the best lemmas, however, GPT-4 could not use it correctly in the first couple of tries. It was only because of our **novel stateful backtracking** approach that we could capture the previous mistakes along with the last error message all in the same prompt which dissuaded the GPT-4 from making repeated mistakes and getting to the right prediction. It is also important to note that we could not solve this problem without retrieval, but once we know the list of relevant lemmas, knowing how to use them becomes extremely important. Given this example, it should not be surprising that even when DPR performs better than BM25, COPRA has an edge because it can act as per the rich feedback it gets from the environment. Another interesting idea one can try is to let COPRA create a search query to control the retrieved lemma, which can make a simple BM25 search more effective than DPR.

---

> > > > ### Comment · Reviewer_Ub88 · 2024-06-06
> > > > **Questions about "stateful backtracking"**
> > > >
> > > > Thank you for your response. I noticed that your paper highlights stateful backtracking as a significant contribution. However, this approach appears similar to traditional planning methods in reinforcement learning. Additionally, recent works like Reflexion and Reasoning as Planning also employ a comparable strategy of using intermediate reasoning steps or checkpoints for self-improvement/planning. Could you clarify how your work differs from these methodologies?

---

> > > > > ### Author Response · Authors · 2024-06-06
> > > > > **Answers to your questions**
> > > > >
> > > > > Thank you for your comment!
> > > > >
> > > > > Yes, like other work on LLM agents, we are inspired by traditional planning/RL. However, the use of LLMs puts us in a very different space in terms of capabilities. Specifically, while traditional symbolic planning and deep RL have been used for theorem-proving in the past, they haven't been able to scale to the kind of problems that Copra can solve.
> > > > >
> > > > > It is true that the idea of LLM agents is also explored in efforts like Reflexion. However, the theorem-proving setting opens up some specific opportunities that the approaches you mention do not exploit. Specifically, formal proofs are driven by a rigorous ground-truth specification (the theorem's statement) and can offer detailed error messages explaining the failures of specific actions, and the Copra agent makes use of these. Also, the stateful backtracking search used in Copra is carefully designed to exploit specific characteristics of the theorem-proving setting (for example, we retain information about actions that failed in the past) and is more sophisticated than the sampling-based search in methods like Reflexion.

---

> > > > > > ### Comment · Reviewer_Ub88 · 2024-06-06
> > > > > > **Thanks for the answers**
> > > > > >
> > > > > > I appreciate the author's effort in addressing my questions. While I still have concerns about the novelty of this paper, their response has alleviated many of my initial reservations. Consequently, I have increased my score

---

### Official Review · Reviewer_G5kH · 2024-05-24

**Rating:** 7
**Confidence:** 4
**Ethics Flag:** 1

**Summary:**

This paper proposes an in-context learning approach for solving the task of automated theorem proving. An ICL agent, called COPRA is built to search for a proof in DFS by repeatedly running the proof step generated by GPT-4-turbo with backtracking.

COPRA has several unique features.
- COPRA filters out and avoid bad proof steps that result in new proof states weaker than old proof states, since we get no progress from these proof steps.
- COPRA fixes runtime errors via self-critique.
- CORPA could retrieve and get benefits from lemmas and definitions and informal texts.
- COPRA could work on both Lean and Coq, two tactic-based proof assistants.

On the Lean miniF2F and Coq CompCert benchmarks, COPRA achieves better performance than the few-shot prompting baseline that targets on generating the full proof in one LLM call. It also performs comparably to the finetuning baselines.

**Questions To Authors:**

1 From the results, retrieval is very important, improving the pass rate from 26.63 to 29.09. Can you explain in more detail what the input and output are in the step of retrieval, and how do you get the output?
2 The numbers of ReProver listed in Table 1 are different from the numbers reported in LeanDojo paper (25 vs 26.5). Did you run ReProver by yourself and get the results?
3 Do you have ICL exemplars in prompts? If so, what is the format of ICL exemplars?
4 Can you provide some statistics for the size of our prompts and LLM responses, like the mean, min, max, per request, and per proof?

**Reasons To Accept:**

Similar to code agents, theorem proving can be tackled by AI agents. This paper brings such a method, which should be beneficial to the AI/TP community.
The proposed proving agent is well-designed, combining the guidance from LLM with the search algorithm. The proposed progress check seems to be a clever and effective approach.
The experiment protocol and evaluation metric are sound and solid. The results are positive to show the power of ICL for theorem proving. Ablation demonstrates the importance of retrieval.

**Reasons To Reject:**

I didn't find major issues for this submission, except for a few questions to be clarified below.

---

> ### Author Rebuttal · Authors · 2024-05-28
>
> Thank you for your thorough feedback. We answer your main questions below.
>
> **Q**: More details about input and output in the step of retrieval.
>
> **A**: The retrieval works by: (i) first tokenizing the proof state, then (ii) performing a BM25 search on the lookup repositories (Mathlib in case of miniF2F) to find the top k lemmas (we use k=7). These lemmas are then added to the prompt along with the proof state. The following example shows the retrieval prompt:
> ```
> Goals to prove:
> [GOALS]
> [GOAL] 1
> n = 70
> [HYPOTHESES] 1
> [HYPOTHESIS] n : ℕ
> [HYPOTHESIS] h₀ : 0 < n
> [HYPOTHESIS] h₁ : n.gcd 40 = 10
> [HYPOTHESIS] h₂ : n.lcm 40 = 280
> [THEOREMS] 1
> [THEOREM] pnat.gcd_mul_lcm : ...
> ...
> [THEOREM] pnat.lcm_coe : ...
> ...
> [INCORRECT STEPS]
> ...
> [STEP] have h₃ : 10 * 280 = n * 40, by rw [←h₁, ←h₂],
> linarith,
> ...
> [LAST STEP]
> rw [←h₁, ←h₂] at h₃,
> [ERROR MESSAGE]
> ....... failed, unknown 'h₃' local
> [END]
> Response:
> [RUN TACTIC]
> have h₃ : n * 40 = 10 * 280, by rw [←nat.gcd_mul_lcm n 40, h₁, h₂],
> linarith,
> [END]
> ```
> **Q**: Did we run ReProver?
>
> **A**: The results we report are those we obtained from running ReProver ourselves. We chose to run ReProver ourselves to obtain more fine-grained information to allow for a comparison of the number of queries and wall-clock time.
>
> **Q**: What is the format of ICL exemplars?
>
> **A**: We use fixed exemplars in all our prompts. The simple examples are intended to demonstrate the prompt format to the LLM. For miniF2F we use 6 examples, these are simple examples with no overlap with the miniF2F test set. The format is shown below:
> ```
> `example_user`
> Goals to prove:
> [GOALS]
> [GOAL] 1
> x = 1 ∧ y = 1
> [HYPOTHESES] 1
> [HYPOTHESIS] x y : ℝ
> [HYPOTHESIS] h₁: x = 3 - 2 * y
> [HYPOTHESIS] h₂: 2 * x - y = 1
> [END]
> `example_assistant`
> [RUN TACTIC]
> linarith,
> [END]
> `example_user`
> Goals to prove:
> [GOALS]
> [GOAL] 1
> x = 1 ∧ y = 1
> [HYPOTHESES] 1
> ...
> [LAST STEP]
> linarith,
> [ERROR MESSAGE]
> linarith failed ...
> [END]
> `example_assistant`
> [RUN TACTIC] split, [END]
> ```
> **Q**: Statistics on the prompt sizes.
>
> **A**: The statistics for the number of tokens in the prompt sizes for some of our settings are as follows: (this is for GPT-4-turbo model which allows at max 128k tokens)
> ```
> Setting         | Min  |  Max  | Avg
> miniF2F         | 33   | 1828  | 322.61
> miniF2F + ret   | 181  | 4355  | 824.21
> miniF2F + inf   | 192  | 4355  | 955.20
> Compcert + ret  | 292  |116574 | 3219.79
> ```
> Our system prompts typically have 2-2.5k tokens -- examples can be found in the Appendix.

---

> > ### Comment · Reviewer_G5kH · 2024-06-05
> >
> > Thank you for the author's response, which addresses my questions.

---

> > > ### Author Response · Authors · 2024-06-05
> > >
> > > Thank you, we are glad to hear that. Please let us know if you have any further questions, we would be happy to answer them.

---

### Decision · Program_Chairs · 2024-07-10

**Decision:**

Accept

**Comment:**

While there was not a complete consensus on the value of the paper, I believe overall its advantages are outweighing the faults it has. I highly recommend the authors to highlight to what extent they rely on GPT4 or the LLM for the proof, and to what extent their method adds value. Overall, good paper, perhaps requires a bit more polishing.